# Pushing redox potentials to highly positive values using inert fluorobenzenes and weakly coordinating anions

Christian Armbruster [1,3], Malte Sellin [1,3], Matthis Seiler [1], Tanja Würz[1], Friederike Oesten[1], Maximilian Schmucker[1], Tabea Sterbak[1], Julia Fischer[1], Valentin Radtke [1], Johannes Hunger [2] ✉ & Ingo Krossing [1] ✉

While the development of weakly coordinating anions (WCAs) received much attention, the progress on weakly coordinating and inert solvents almost stagnated. Here we study the effect of strategic F-substitution on the solvent properties of fluorobenzenes $C_6F_xH_{6-x}$ (xFB, x = 1–5). Asymmetric fluorination leads to dielectric constants as high as 22.1 for 3FB that exceeds acetone (20.7). Combined with the WCAs $[Al(OR^F)_4]^-$ or $[(^FRO)_3Al\text{-}F\text{-}Al(OR^F)_3]^-$ ($R^F = C(CF_3)_3$), the xFB solvents push the potentials of $Ag^+$ and $NO^+$ ions to +1.50/+1.52 V vs. $Fc^+/Fc$. The xFB/WCA-system has electrochemical xFB stability windows that exceed 5 V for all xFBs with positive upper limits between +1.82 V (1FB) and +2.67 V (5FB) vs. $Fc^+/Fc$. High-level ab initio calculations with inclusion of solvation energies show that these high potentials result from weak interactions of the ions with solvent and counterion. To access the available positive xFB potential range with stable reagents, the innocent deelectronator salts $[anthracene^F]^{+}[WCA]^-$ and $[phenanthrene^F]^{+}[WCA]^-$ with potentials of +1.47 and +1.89 V vs. $Fc^+/Fc$ are introduced.

The elementary steps underlying the reversible addition and removal of electrons from matter−metals, molecules or materials−are the basis of redox chemistry, electrocatalysis, and electrochemical energy storage. Since the electrochemical potentials of such reactions are only comparable within one solvent, the IUPAC suggests the use of the ferrocenium/ferrocene redox couple $Fc^+/Fc$ as a reference[1,2]. Due to the weak solvent dependence of its potential, other electrochemical potentials can be compared for various liquid media (Ferrocene Assumption)[3].

By contrast to the multitude of reagents for electronation (addition of an $e^-$) down to the potential level of solvated electrons, accessible and reliable reagents for deelectronation[4] (removal of an $e^-$) at potentials higher than about +0.5 to +1 V vs. $Fc^+/Fc$ are scarce, but would be useful[5]. Note that we use electronation[4] and deelectronation in their strict sense, i.e., addition and removal of electrons in innocent

reactions, respectively, hence without non-innocent complications such as complex formation, substitution or degradation reactions. For reasoning, see the reports by Radtke[6] and Himmel[7] et al. The typical systems available, for example, salts of the inorganic ions $NO^+$ and $Ag^+$, are applied in many fields of chemistry (organic synthesis[8–10], nitrosonium complexes[11]) and the material sciences (batteries[12], solar cells[13], silver electrode[14–16]) to remove electrons from the system studied. Such commercial deelectronator salts include classical, but rather small weakly coordinating anions (WCAs), like $[BF_4]^-$ or $[MF_6]^-$ (M = P, As, Sb)[17]. Yet, solutions of such salts $de^+[WCA]^-$ ($de^+ = Ag^+$, $NO^+$) show a strong solvent (S) dependence[5] of their potentials vs. $Fc^+/Fc$, ranging for $NO^+$ from +0.56 (S = DMF) to +1.00 V (S = $CH_2Cl_2$) and for $Ag^+$ from −0.07 (S = MeCN) to +0.65 V (S = $CH_2Cl_2$)[5,18–21]. This potential dependence arises from partially strong interactions of the $NO^+$ or $Ag^+$ cations with the solvent molecules S. In addition, especially the

[1]Institut für Anorganische und Analytische Chemie and Freiburger Materialforschungszentrum (FMF), Albert-Ludwigs-Universität Freiburg, Albertstr. 21, 79104 Freiburg, Germany. [2]Molecular Spectroscopy Department, Max-Planck-Institut für Polymer Research, Ackermannweg 10, 55128 Mainz, Germany. [3]These authors contributed equally: Christian Armbruster, Malte Sellin. ✉e-mail: hunger@mpip-mainz.mpg.de; krossing@uni-freiburg.de

systems with the highest de[+] potentials—published in weakly basic and little interacting low-polarity organic media like CH$_2$Cl$_2$—suffer from the low solubility of the salts de[+][WCA][-]. This leads to reduced activities $a$(Ag[+], S) or $a$(NO[+], S) that may further be lowered by severe ion-pairing of de[+][WCA][-] in S. Equally important, the de[+] ions form rather strong bonds to classical aromatic solvents[19,22] that greatly lower the accessible potential. Moreover, solvated NO[+] is only fleetingly stable in arenes, as the formed Wheland complex[23,24] serves as an entry point to an electrophilic aromatic substitution reaction[18,19].

Fundamentally, both problems—ion-pairing and solvent coordination, substitution, or degradation—may be overcome by using the salt de[+][WCA][-] in a combination of an almost non-interacting inert, but polar and redox-robust (aromatic) solvent with a redox-stable very good WCA. Despite being polar with a large dipole moment, the solvents have to act as weak bases (thermodynamics) and nucleophiles (kinetics). Together with large WCAs, such solvents enhance the solubility of de[+][WCA][-] by solvating de[+] and [WCA][-] and yield dissociated ions that have high activities $a$(de[+], S) also in low dielectric media S (Fig. 1).

Our entry to the field was the preparation of the NO[+] and Ag[+] salts of the large WCAs [*pf*][-] (= [Al(OR$^F$)$_4$][-])[17,25] and [*al-f-al*][-] (= [($^f$RO)$_3$Al-F-Al(OR$^F$)$_3$][-]; R$^F$ = C(CF$_3$)$_3$)[17,26] (Fig. 1) that enable de[+][WCA][-] solubility in low-polarity solvents, i.e., CH$_2$Cl$_2$, 1FB and 5FB, but also the more polar 2FB, 3FB and 4FB (Fig. 1). We routinely use these WCA-reagents for syntheses with Ag[+27–29] or NO[+] ions[30–35], for deelectronation[34–42] or dehalogenation reactions[43,44]. More than 100 groups worldwide utilize the advantageous properties of [*pf*][-] and closely related WCAs for applications in diverse fields, i.e., deelectronation[45], supporting[46–48] and battery[49–53] electrolytes, organic solar cells[54–56], photoacids[57], catalysis[58], etc.[59–61].

This account reports how the above ion-pairing and solvent problems upon application of de[+] salts are overcome by using strategically substituted and dipole-maximized polar fluorobenzenes[62] xFB (x = 1–5) partnered with salts of the large[63] redox-stable WCAs [*pf*][-] and [*al-f-al*][-] shown in Fig. 1. Especially the higher fluorinated 4FB and 5FB solvents are very weak ligands and induce, together with the aluminate WCAs, record high de[+]-potentials in xFB solution of up to +1.52 V for NO[+](4FB)/NO and +1.50 V Ag[+](5FB)/Ag, both vs. Fc[+]/Fc in S. In addition, problems with reported literature potentials in standard solvents were spotted[19], addressed and corrected in this work[5,18]. Two innocent

deelectronator reagents push the positive potential accessible in xFB solution to synthetically highly useful positive values of up to +1.89 V vs. Fc[+]/Fc. Both overcome the observed non-innocent reactivity of the Ag[+] or NO[+] ion[33,64–66] and make use of the very large electrochemical xFB stability windows that exceeds 5 V at positive upper limits between +1.82 V (1FB) and +2.67 V (5FB) vs. Fc[+]/Fc.

## Results

First, we describe the missing solvent properties of fluorobenzenes, before investigating the de[+] potentials in standard and xFB solution with large WCAs, validate and relate the measured potentials to electrolyte species with crystal structures, IR spectra and high-level quantum chemical calculations. Finally, we utilized these strategies to even further push the positive potential limits with perfluorinated arenium cations.

### Permittivity and principal solvent properties of xFBs

To understand the effect of the solvent, we determined fundamental xFB (x = 1–6) properties, such as dielectric constants, dipole moments, viscosities, and densities. The static dielectric constant $\varepsilon_s$ is a measure for the dipole density and polarizability within a solvent S. Hence, it governs the stabilization of any ion in the solvent and is also used as a polarity descriptor for many quantum chemical solvation energy calculations. Dielectric relaxation spectroscopy was used to measure $\varepsilon_s$, the relaxation time, $\tau$, and the infinite frequency permittivity, $\varepsilon_\infty$, of the neat xFB solvents as well as their binary 1:1 mixtures between 5 and 30 °C. From the dielectric relaxation strengths $\varepsilon_s$-$\varepsilon_\infty$, we derive[67] the effective dipole moment, $\mu_{eff}$, of xFB in the liquid phase (Supplementary Note 5). For completion, the in part unknown densities, kinematic/dynamic viscosities of all xFB solvents were measured with an Ubbelohde-viscometer at 23.1 ± 0.2 °C and were added to Table 1 (cf. Supplementary Note 9).

**Pure xFB solvents.** The $\varepsilon_s$ values of neat xFB at 25 °C in Table 1 agree, where known (1–2FB, 5–6FB)[68,69], within ±0.5 units to the literature values measured between 22 and 25 °C. $\varepsilon_s$(xFB) increases with increasing fluorine content from 1FB to 3FB, but decreases upon further fluorination. As $\varepsilon_s$ is largely determined by the alignment of molecular dipoles according to an external electric field, $\varepsilon_s$(xFB) mostly reflects the trends of the molecular dipole moments of xFB, and the $\mu_{eff}$(xFB)[70] values in Table 1 agree well with the values for isolated xFB molecules obtained from DFT calculations. They are only moderately affected by a slight preferential parallel alignment of xFB's molecular dipoles in the liquid phase (Supplementary Note 9).

$\varepsilon_s$ decreases with increasing temperature for all xFB solvents, consistent with thermal motion countering the correlations of the molecular dipoles according to the external field. The slopes of $\varepsilon_s$ vs.

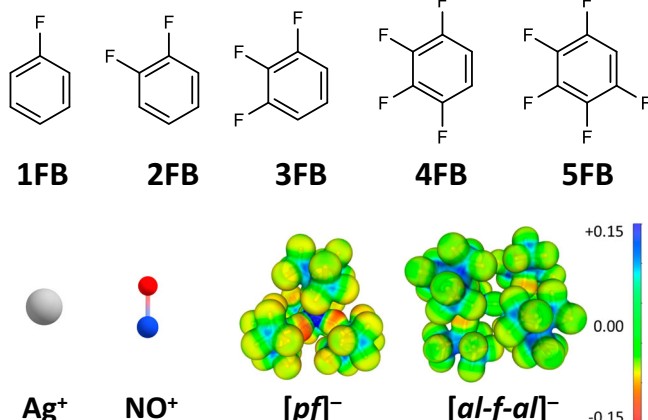

**Fig. 1 | Summary of the investigated solvents, cations and anions.** Top: Structural formulae of the strategically substituted and dipole-maximized polar fluorobenzenes xFB with x = 1–5. Bottom left: Ball and Stick models of the Ag[+] and NO[+] cations. Bottom right: The WCAs [*pf*][-] (= [Al(OR$^f$)$_4$][-]) and [*al-f-al*][-] (= [($^f$RO)$_3$Al-F-Al(OR$^F$)$_3$][-]; R$^F$ = C(CF$_3$)$_3$) in a representation with the electrostatic potential plotted on an isodensity surface (0.025 $e^-$ B$^{-3}$) using like cut-off values and calculated at the (RI-)BP86/def2-TZVPP level of theory. The scale bar right indicates areas of more negative (red), neutral (green) and positive (blue) surface potentials.

**Table 1 | Relaxation parameters of the neat solvents at 25 °C: static dielectric constant $\varepsilon_s$, infinite frequency permittivity $\varepsilon_\infty$, relaxation time $\tau$, and the derived effective dipole moments $\mu_{eff}$. Kinematic ($\nu$) and dynamic ($\eta$) viscosities at 23.1 ± 0.2 °C as well as densities $\rho$ of xFB at the indicated temperatures [T]**

| Solvent | $\varepsilon_s$ | $\varepsilon_\infty$ | $\tau$/ps | $\mu_{eff}$/Debye | $\nu^a$/mm$^2$ s$^{-1}$ | $\eta^b$/mPa•s | $\rho$ [T]/g cm$^{-3}$ [°C] |
|---------|------|------|------|------|--------|--------|--------|
| **1FB** | 5.8  | 2.4  | 6.4  | 1.9  | 0.5540 | 0.5662 | 1.022 [23] |
| **2FB** | 14.2 | 2.5  | 10.2 | 3.6  | 0.5378 | 0.6217 | 1.156 [23] |
| **3FB** | 22.1 | 2.6  | 16.9 | 4.7  | 0.5389 | 0.6962 | 1.292 [23] |
| **4FB** | 12.6 | 2.3  | 14.7 | 3.5  | 0.5586 | 0.7826 | 1.401 [23] |
| **5FB** | 4.5  | 2.1  | 11.2 | 1.8  | 0.5403 | 0.8165 | 1.511 [24] |
| **6FB** | 2.1  | -    | -    | -    | 0.5592 | 0.8982 | 1.606 [25] |

$^a$Uncertainty for all measurements: ±0.0009 mm$^2$ s$^{-1}$.
$^b$Uncertainty for the measurements between ±0.0010...0.0015 mPa•s.

temperature correlate with $\mu_{eff}$ and are steepest for the most polar 3FB, 2FB and 4FB (Supplementary Fig. 101). Cooling increases $\varepsilon_s$(3FB) considerably to 24.6 at 5 °C, which is 2.5 units higher than at 25 °C. This may be used in favor for synthetic purposes.

**Binary 1:1 xFB mixtures.** The trends for the binary 1:1 mixtures discussed in Supplementary Note 5 parallel those found for the neat solvents: With increasing fluorine content of one component mixed with another xFB, we find $\varepsilon_s$ to peak for mixtures with 3FB, irrespective of the second component (Supplementary Figs. 105b–109b). Hence, binary mixtures of xFB's yield solvents with a tunable permittivity between $\varepsilon_s$ = 2–22 at 25 °C, which can be exploited in specific applications. The moderate parallel correlation of the dipoles observed for the neat xFBs are reduced in the mixtures, as apparent from the effective dipole moments $\mu_{eff}$ (Supplementary Fig. 110). All binary 1:1 mixtures also show a monotonic increase of $\varepsilon_s$ with decreasing temperature (Supplementary Fig. 111).

**Comparison of $\varepsilon_s$(xFB) to standard solvents.** The high static dielectric constant of 3FB of $\varepsilon_s$ = 22.1 stems from the very high dipole moment of $\mu_{eff}$ = 4.7 D. It even exceeds the room temperature value of the prototypical polar, but coordinating solvent acetone (20.7) and by far exceeds that of the typically for synthesis used aprotic non-basic, mainly non-coordinating media like $CH_2Cl_2$ (8.9) or 1,2-$Cl_2C_2H_4$ (10.8)[68]. In addition, also non-basic, but polar liquid sulfur dioxide (13.8 at 25 °C), e.g., used to stabilize carbocations and many other reactive cations, is surpassed[71]. We note that $\varepsilon_s$ of 2FB (14.2) and 4FB (12.6) also exceed those of $CH_2Cl_2$ and 1,2-$Cl_2C_2H_4$ and reach that of $SO_2$. In addition, $\varepsilon_s$ of 1FB and 5FB are similar in magnitude, and it appears that also the affordable 5FB is a hitherto underestimated non-basic solvent for reactions that need some polarity, i.e., reaching that of the frequently used, but toxic $HCCl_3$ (4.8) or chlorobenzene (5.6). To fully understand the solvent's effect on ionic solutes—besides considering ionic species being stabilized by the solvent as an effective medium with dielectric permittivity $\varepsilon_s$—also specific interactions between the solvent and the ionic species have to be considered. Such explicit xFB-solvation is discussed in the following sections.

**Cyclovoltammetric measurements in xFB and standard solvents**
**Demand for reevaluation of de$^+$ potentials in standard solvents.** The published[5] formal potentials of the Ag$^+$(S)/Ag system in acetonitrile (S = AN) and dichloromethane (S = $CH_2Cl_2$) are by 0.61 V markedly different. In this light, the close similarity of the potentials of the NO$^+$(S)/NO system in AN (+0.87 V) and $CH_2Cl_2$ (+1.00 V) was somewhat surprising[5,18,19]. Apparently, these opposing solvent effects are due to experimental challenges when measuring the potentials that are caused by the very low solubility of NO$^+$[BF$_4$]$^-$ in $CH_2Cl_2$[19]. Therefore, Kochi et al.[19] used NO(g) dissolved in $CH_2Cl_2$ for the measurements. Yet, in a footnote, the authors honestly stated that they were unable to accurately prepare solutions of gaseous NO with known concentrations[19]. This suggests this potential value to be considerably too low due to the very low activities $a$(NO$^+$, $CH_2Cl_2$) accessible with [BF$_4$]$^-$ counterion. In addition, several formal potential values of de$^+$ ions included in the highly-cited 1996 Geiger-Conelly review[5] are given as estimates. Similar to the NO$^+$[BF$_4$]$^-$ case above, some of these values may be drastically affected by ion-pairing. In the course of our investigations, we realized that xFB solutions of NO$^+$[pf]$^-$ or NO$^+$[al-f-al]$^-$, although being colored due to the formation of Wheland intermediates (see below)[37], are stable for many days at room temperature. Hence, a comprehensive (re-)investigation of the de$^+$ potentials in standard as well as fluorobenzene solvents appeared timely.

**Methodology for CV.** With the help of a large WCA, we study the principal magnitudes decisive for any possible use of a solvent in electrochemistry. Thus, using 100 mM [NBu$_4$]$^+$[pf]$^-$ as supporting

electrolyte in the solvent S, the negative potential limit $E_{neg}$, the positive potential limit $E_{pos}$ and the resulting widths of the potential stability windows (ECW) of xFB's and several of the above stated standard solvents were determined. Note that other large WCA salts including [B(C$_6$F$_5$)$_4$]$^-$ or [B(Ar$^F$)$_4$]$^-$ (Ar$^F$ = 3,5-(CF$_3$)$_2$C$_6$H$_3$) and that were recommended to be used as supporting electrolytes especially in low permittivity media[72–75], reduce the ECW due to decomposition of the anions at higher potentials: Waldvogel et al. demonstrated that both WCAs decompose at anodic limit potentials[76] of only +1.25 to 1.78 V vs. Fc$^+$/Fc in AN in electrosynthetic yields with formation of the respective fluorinated biphenyls. Hence, the potentials of Ag$^+$, NO$^+$ and in addition that of the organic amine TBPA$^0$ (TBPA$^0$ = N(4-Br-C$_6$H$_4$)$_3$)[5], supposed to be less affected by medium effects, were investigated vs. Fc$^+$/Fc. The general CV-measurement setup is described in Supplementary Note 3.1. Full results and all CV traces at scan rates between 20 and 200 mV s$^{-1}$ are included in Supplementary Notes 3.2 to 3.7 of the Supplementary Information.

The values of the negative limit $E_{neg}$, the positive limit $E_{pos}$ and the resulting ECW-widths of xFB and standard solvents (Supplementary Note 3.3) are collected in Table 2, known literature values in Supplementary Table 6.

**Comparison to literature data for standard solvents.** Our ECW-widths of the standard solvents measured with the supporting electrolyte [NBu$_4$]$^+$[pf]$^-$ in Table 2 are much larger than the corresponding values found in the literature (Supplementary Table 6). These differences are probably induced by the use of different supporting electrolytes (often [NBu$_4$]$^+$[ClO$_4$]$^-$ or [NBu$_4$]$^+$[PF$_6$]$^-$) that influence the ECW. In part, a standard calomel electrode (SCE) was used as a reference electrode (RE) including a Liquid Junction Potential (LJP) between the aqueous RE and the non-aqueous solvent[6,21]. Further, in some cases, the supporting electrolyte used was not given and prevents a more accurate comparison, although measured with the same RE as in our work. Nevertheless, the literature values help put our ECWs into context. Among that, the used electrolyte salt is key for the ECW: Own experiments in $CH_2Cl_2$ and using either [NBu$_4$]$^+$[pf]$^-$ or [NBu$_4$]$^+$[PF$_6$]$^-$ as supporting electrolyte show that the [pf]$^-$ salt gives an ECW which is by about 1 V wider than that of the [PF$_6$]$^-$ salt (Table 2, Supplementary Fig. 26). The larger window, which is induced by the [NBu$_4$]$^+$[pf]$^-$ supporting electrolyte with reduced ion-pairing, is a favorable feature that can be exploited for further developments in the field of electrochemistry in general.

**Performance of the xFB solvents.** The $E_{pos}$ potential shows a progression from 1FB to 5FB and increases from +1.82 V (1FB) up to +2.67 V (5FB), consistent with the rising ionization energies (IEs) of these solvents with increasing fluorination (Table 2). The $E_{neg}$ potentials of 1-4FB are rather similar and around −3.1 ± 0.1 V. Only the most fluorinated 5FB shows the highest $E_{neg}$ potential at −2.37 V, as expected due to its pronounced[77] C-H acidity. Altogether these potentials lead to ECWs exceeding 5 V that peak for 4FB at 5.51 V. They are compatible with a wide range of electrochemical syntheses. Hence, together with the poor capacity of the xFB solvent molecules to serve as ligands (see also the following sections), the basic electrochemical performance of xFB with [NBu$_4$]$^+$[pf]$^-$ as supporting electrolyte salt is very promising.

After having established the ECW-widths, we proceeded to determine the potentials of the electroactive salts de$^+$[pf]$^-$ in these solvents and the effect of the—standard and xFB—solvent on their potentials. Our measurements of the redox potentials $E$ of 10 mM solutions of the cations de$^+$ = NO$^+$ and Ag$^+$ with [NBu$_4$]$^+$[pf]$^-$ as supporting electrolyte in the solvents xFB, $CH_2Cl_2$, 1,2-$Cl_2C_2H_4$, dimethylformamide (DMF) and AN are collected in Table 2 and, for the standard solvents, compared to published data. Where appropriate, $E_{1/2}$ values were converted into formal potentials $E^{o'} = E_{1/2}$, i.e., for the Ag$^+$/Ag and the Fc$^+$/Fc systems. This is justified, if the diffusion

**Table 2 | Electrochemical properties of selected solvents S and redox active species therein**

| Solvent S | $E_{neg}$ | $E_{pos}$ | ECW | $IE$ / eV | $E_{1/2}(NO^+(S)/NO)$ | $E°'$ $(Ag^+(S)/Ag)$ |
|---|---|---|---|---|---|---|
| 1FB | −3.18 | +1.82 | 5.00 | 9.20[68] | +1.11 | +0.74 |
| 2FB | −3.09 | +2.05 | 5.14 | 9.29[68] | +1.23 | +0.99 |
| 3FB | −3.02 | +2.35 | 5.37 | 9.40[118] | +1.42 | +1.26 |
| 4FB | −3.08 | +2.43 | 5.51 | 9.53[68] | +1.52 | +1.47 |
| 5FB | −2.37 | +2.67 | 5.04 | 9.63[68] | +1.47 (+1.47 with [al-f-al]⁻) | +1.50 |
| $CH_2Cl_2$ | −2.91 (−2.62[a]) | +3.36 (+2.58[a]) | 6.27 (5.20[a]) | 11.33[b] | +1.40 [+1.00[c,d]] (+0.95[e]) | +0.88 [+0.65[d]] |
| 1,2-$Cl_2C_2H_4$ | −2.28 | +3.13 | 5.41 | 11.07[b] | +1.23 | +0.66 |
| DMF | −3.60 | +1.73 | 5.33 | 9.13[b] | +0.54 [+0.56[c]] | +0.11 [+0.49[c,f]] |
| AN | −3.09 | +4.59 | 7.68 | 12.20[b] | +0.87 [+0.87[c]] | +0.02 [+0.06[g]] |
| PC | −3.50 | +4.15 | 7.65 | (10.5)[b,c] | | |
| THF | −3.60 | +1.33 | 4.93 | 9.40[b] | | |

Negative reduction $E_{neg}$, positive oxidation $E_{pos}$ limits and derived potential windows of stability (ECW) were derived from CV measurements of the selected solvents at $v = 100$ mV s⁻¹ with 100 mM [NBu₄]⁺[$pf$]⁻ as supporting electrolyte in S, if not stated otherwise. For comparison, the Ionization Energies $IE$ of the solvents are included. Cyclovoltammetrically determined potential values of the redox systems Ag⁺(S)/Ag and NO⁺(S)/NO, with S as xFB (x = 1–5) or other solvents are given. The CVs were obtained with Pt disc electrodes, the used counterion was [$pf$]⁻, the supporting electrolyte was 100 mM [NBu₄]⁺[$pf$]⁻, the scan rate was varied between 20 and 200 mV s⁻¹, but $E_{1/2}$ is independent of the rate. Literature values are given in square brackets in italics where available, own control experiments with alternative reagents are given in parentheses in regular font. All potential values are given in V with reference to $E°'$((Fc⁺/Fc) S) with an uncertainty of ±0.025 V. All measurement data are given in the Supplementary Information (xFB: NO⁺/NO Supplementary Notes 3.4.1 (with respect to Fc⁺/Fc) and 3.5.2 (with respect to Ag⁺/Ag); Ag⁺/Ag Supplementary Note 3.5.1. Other solvents: NO⁺/NO Supplementary Note (with respect to Fc⁺/Fc) and 3.7.2 (with respect to Ag⁺/Ag); Ag⁺/Ag Supplementary Note 3.7.1).
[a][NBu₄]⁺[PF₆]⁻ as conducting salt.
[b]NIST Chemistry Webbook https://webbook.nist.gov/chemistry/.
[c]Appearance energy from [b]. Values from the literature[5].
[d]From the literature[5,19], using Ag⁺[PF₆]⁻ with [NBu₄]⁺[PF₆]⁻ and NO⁺[BF₄]⁻ with [NBu₄]⁺[BF₄]⁻ as supporting electrolyte.
[e]Independent own control measurement in CH₂Cl₂ using 10 mM NO⁺[PF₆]⁻ with 100 mM [NBu₄]⁺[PF₆]⁻ as supporting electrolyte (See Supplementary Note 3.6.1.3).
[f]This potential was given as an estimate in ref. 5 and appears to be completely wrong. Our value is confirmed in the triangular evaluation in Supplementary Note 3.7.3.
[g]Value from the literature[21].

coefficients of the electronated and deelectronated form of the redox system are identical. For the Fc⁺/Fc system, this was verified to be almost true in many solvents S by diffusion constant PFGSE-NMR measurements. Hence, from the measured diffusion constants, $E°'$ and $E_{1/2}$ were calculated to be similar within about 10 mV in 1FB and 5FB, and less for the other solvents (Supplementary Note 3.2.1). However, for the NO⁺/NO system, where NO⁺ forms a Wheland complex [NO(xFB)]⁺ (see below), which diffuses much slower than a neutral NO molecule, we give $E_{1/2}(NO^+(S)/NO)$. In addition, we measured the potentials of the deelectronation of 10 mM solutions of neutral TBPA⁰ giving the frequently used radical cation [TBPA]⁺, which is well-known from the *magic blue* salt (Supplementary Note 3.4.2)[5,78]. Using the neutral underlying the radical cation [TBPA]⁺ with the [$pf$]⁻ counterion from the electrolyte solution prevents electrochemical side reactions of the [SbCl₆]⁻ anion, compared to using the *magic blue* salt directly. Exemplarily we display a stack plot of the measurements of Fc⁺[$pf$]⁻ solutions in xFB and CH₂Cl₂ and with respect to the Ag⁺(S)/Ag redox system in Fig. 2.

Note that the systems are electrochemically quasi-reversible since the peak-to-peak separation of the waves in the CVs increases with the scan rate. Exemplarily, the $E_{1/2}$ potential of a 10 mM solution of the salt NO⁺[al-f-al]⁻ with the least coordinating, largest WCA was measured in the least polar and coordinating solvent 5FB. The potential influence of the counterion and possible ion-pairing effects are expected to be most prominent in this solvent. Yet, we find identical $E_{1/2}(NO^+(5FB)/NO)$ values for both WCAs (+1.47 V), suggesting that the measured potential values are insensitive to the WCA counterion even in low-polarity 5FB.

**Potentials of the de⁺/de systems in standard solvents S.** The Ag⁺(CH₂Cl₂)/Ag and NO⁺(CH₂Cl₂)/NO potentials determined with [$pf$]⁻ WCA in Table 2 differ from the published values by +0.23 V (Ag⁺ with [PF₆]⁻ counterion) and +0.40 V (NO⁺ with [BF₄]⁻ counterion). By contrast, the NO⁺(S)/NO potentials in S = DMF; AN and the Ag⁺(AN)/Ag potential agree within 0.04 V with literature values. Apparently, DMF and AN are polar and coordinating enough so that ion-pairing plays a minimal role, and the potentials are independent of the counterion. To

prove this point, we measured the NO⁺(CH₂Cl₂)/NO potential using 10 mM NO⁺[PF₆]⁻ with 100 mM [NBu₄]⁺[PF₆]⁻ as supporting electrolyte. This gave $E_{1/2}$ as +0.95 V (Supplementary Note 3.6.1.3), close to the published value of +1.00 V[19]. But note, in this work[19] CH₂Cl₂ was saturated with an unknown concentration of NO gas, which may account for some differences[19]. Therefore, it can be assumed that the [BF₄]⁻ as well as [PF₆]⁻ anions form a strongly bound contact-ion pair in solution. Due to the rather small amount of the solvated free NO⁺ cation, its activity and hence its potential can be markedly lowered. By contrast, the NO⁺ solution of the corresponding [$pf$]⁻ salt in CH₂Cl₂ apparently provides activities unaffected by ion-pairing. Hence, the interaction with the [PF₆]⁻ anion appears to reduce the NO⁺-activity in solution by

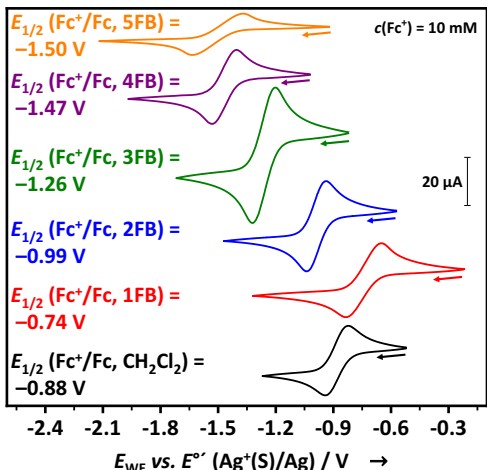

**Fig. 2 | Formal potentials of the Ag⁺ cation in different xFB solvents.** 2nd cycles of the CVs ($v = 100$ mV s⁻¹) of solutions of Fc⁺[$pf$]⁻ ($c = 10$ mM) in the solvents xFB (x = 1–5) and CH₂Cl₂ with the conducting salt [NBu₄]⁺[$pf$]⁻ ($c = 100$ mM) corrected to the formal potential $E°'$ by −0.118 V (cf. Supplementary Note 3.2.1).

about 7 orders of magnitude corresponding to a potential drop of 7•0.059 V = 0.413 V, assuming ideal Nernst'ian behavior. A similar consideration accounts for the 0.23 V difference between the $Ag^+$ potentials in $CH_2Cl_2$: Again, the small $[PF_6]^-$ anion induces strong ion-pairing and lowers the activity of $Ag^+$ ions by four orders of magnitude, if compared to the corresponding $[pf]^-$ salt in $CH_2Cl_2$. Thus, the advantage of the $[pf]^-$ WCA as counterion to provide largely improved activities $a(de^+, S)$ of the electroactive ion in less polar and less coordinating solvents S like $CH_2Cl_2$ becomes very evident.

**Potentials of the $de^+/de$-systems in xFB solution (de = Ag, NO).** For $E(de^+/de, xFB)$, we find a clear increase in the potentials from 1FB to 4FB. From 4FB to 5FB, the $NO^+(xFB)/NO$ potential experiences a slight reduction, while that of $Ag^+(xFB)/Ag$ further increases. Very high potentials of up to +1.52 V ($NO^+(4FB)/NO$) and +1.50 V ($Ag^+(5FB)/Ag$) vs. $E°´(Fc^+/Fc, S)$ are reached. In addition, by appropriate choice of the solvent, the $NO^+(S)/NO$ potential can be tuned by 0.41 V and the $Ag^+(S)/Ag$ potential even by 0.75 V. Apparently, this variation is induced by the reduced interaction energies of the Wheland complexes $[de(xFB)]^+_{solv}$ upon increasing the degree of fluorination of xFB. This complies with the $IE$s collected in Table 2 and the color of the $NO^+$ solution (see below, also Supplementary Note 3.5.4).

**Validation with Born-Fajans-Haber-Cycles.** From the measured $E_{1/2}$ potentials of 10 mM solutions of $Fc^+$ and $NO^+$ versus $E(Ag^+(10$ mM, s)/Ag) and $NO^+$ versus $E(Fc^+(10$ mM)/Fc(10 mM), S) within the same solvent S, triangular Born-Fajans-Haber-Cycles (BFHC) can be constructed. Knowing two out of the three values in the BFHC, the third can be calculated (shown for 5FB and $1,2\text{-}Cl_2C_2H_4$ in the method section, and all data in the Supplementary Notes 3.5.3 and 3.7.3). With this relation, the internal consistency of all the measured data for nine solvents was checked by calculating mean $|\Delta E|$ and $|\Delta G|$ uncertainties that are lower than 0.05 V or 5 kJ mol$^{-1}$ toward the absolute value. Hence, our measurements are consistent within this margin.

**Interaction of $de^+[pf]^-$ with the solvents xFBs, $CH_2Cl_2$ and $1,2\text{-}Cl_2C_2H_4$**
The variation of the deelectronation potentials with solvent and the high values in xFBs suggest that despite their polarity, the direct coordinative interaction capacities of the xFB solvents with the electroactive $de^+$ ions depend on the degree of fluorination of xFB and may be weak. To explore the coordination capacities of the solvents, we investigated their solvated single-crystal X-ray diffraction (scXRD) structures (mainly $Ag^+$-complexes) and IR spectra ($NO^+$-salts). Later this information is discussed in the context of their energetics.

**Silver salts.** The soft silver cation $Ag^+$ is rather carbophilic[40] and a large number of solvent complexes $[Ag(arene)_n]^+$ are published, including a very systematic investigation[22] of the reference system $[Ag(arene)_3]^+[B(C_6F_5)_4]^-$ with the very good WCA $[B(C_6F_5)_4]^-$. Yet, only a few $[Ag(xFB)_n]^+$ salts are known[26,79], the majority of which were published by our group as $[pf]^-$ or $[al\text{-}f\text{-}al]^-$ salts. For S = $CH_2Cl_2$, the types $[Ag(S)_3]^+$ and $[Ag(S)_4]^+$ are known[80,81].

We note that the structural type $[Ag(S)_1]^+[pf]^-$ with solvent-separated cations and anions is absent. Rather, tight ion-pairs $\{Ag[pf]\}_{ip}$ and $\{(S)Ag[pf]\}_{ip}$ are formed. The most relevant molecular (cation) structures of the structural types were collected and exemplarily compiled in Fig. 3.

**scXRD $NO^+$ salts.** We succeeded to crystallize $[NO(2FB)]^+[pf]^-$, thus establishing a $\eta^6$-$NO^+$-Wheland complex with a fluorinated arene (Fig. 4A). Hitherto, only $NO^+$-Wheland complexes of electron-rich arenes (e.g., (alkyl-)benzenes) were crystallized. The electron-deficiency of 2FB lengthens the N···centroid distance in the $[NO(2FB)]^+$ complex (2.255(9) Å) by more than 0.1 Å compared to the

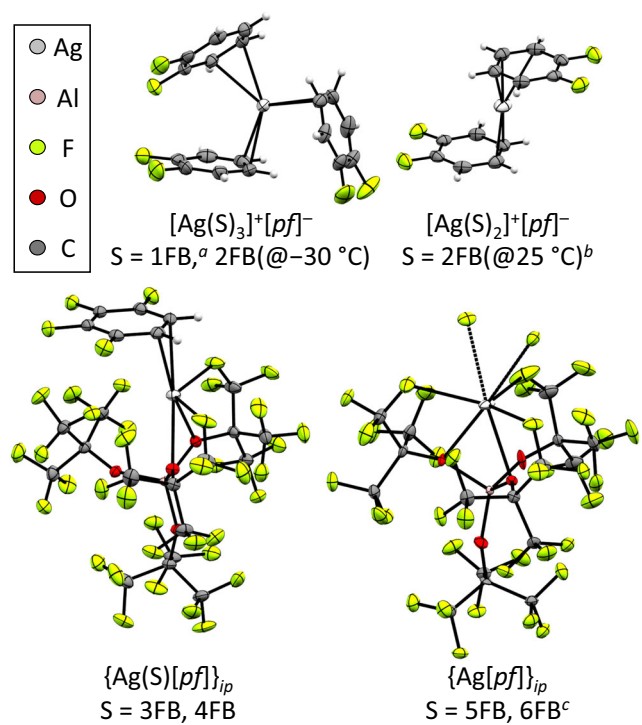

**Fig. 3 | Coordination environments of the $Ag^+$ cation in different xFB solvents.** Exemplarily selected molecular (cation) structures of the most important structural types. For salt structures, the counterion $[pf]^-$ was omitted. In the tight ion-pair $\{Ag[pf]\}_{ip}$, three further contacts to F-atoms of other $[pf]$-units in the unit cell are included. Thermal ellipsoids set at 50% probability. $a$: ref. 79 $b$: ref. 17 $c$: The structure of $\{Ag[pf]\}_{ip}$ crystallized from 5FB and 6FB is isotypic to that obtained earlier from isoperfluorohexane[29].

average N···centroid distances of known 1:1 adducts[82–84]. In addition, the C(F) carbon atoms have longer $NO^+$ contacts of 2.759(9)–2.794(7) Å than their C(H) counterparts (2.49(1)–2.69(1) Å). The O-N···centroid angle, herein 164.8°, is highly flexible in $NO^+$-Wheland complexes due to the π-type interaction from the highly delocalized HOMOs of the arene toward the π* LUMOs of the $NO^+$. It is bent toward those C(H) atoms with large coefficients of the HOMO.

Overall, the scXRD structures are indicative of weaker $Ag^+$–xFB coordination for increasing x, and also, 2FB is weaker bound in the $NO^+$-Wheland-complex than in non-fluorinated counterparts.

**IR-data NO-stretches.** The π-donor properties of different arenes in Wheland complexes with the nitrosonium cation were extensively studied in the 1990s by IR spectroscopy probing the redshift of the $v_{NO}$ vibration relative to that of free $NO^+$ (2340 cm$^{-1}$ in $NO^+[PF_6]^-$)[82]. While $v_{NO}$ of $NO^+[PF_6]^-$ is shifted to 2075 cm$^{-1}$ in benzene, the redshift is significantly larger in hexamethylbenzene with $v_{NO}$ at 1885 cm$^{-1}$. Extending this series, we measured ATR-IR spectra of concentrated $NO^+[pf]^-$ solutions in xFB and observed broad $v_{NO}$ bands that increase from 2007 to 2049 cm$^{-1}$ in 1FB to 3FB (Supplementary Fig. 113). Unexpectedly, the bands of $NO^+[pf]^-$ dissolved in xFB (x = 1–3) are redshifted, if compared to the $NO^+[PF_6]^-$ complex with benzene. This may result through interaction of the Wheland complex with the small $[PF_6]^-$ counterion, i.e., the formation of the ion-pair $\{(C_6H_6)NO[PF_6]\}_{ip}$ reducing the interaction strengths of $NO^+$ with benzene and increasing $v_{NO}$. In line with the strong decolorization of the $NO^+[pf]^-$ solution in going from 3FB to 4FB (Fig. 4E), $v_{NO}$ was not visible in 4–6FB. Due to the low intensity of the band $v_{NO}$ in non-coordinated $NO^+$, this stretching band was invisible in the solution. In addition, spectra of the evaporated residue of $NO^+[pf]^-$ solutions in xFB were measured directly on the ATR-IR unit inside a glovebox to investigate the

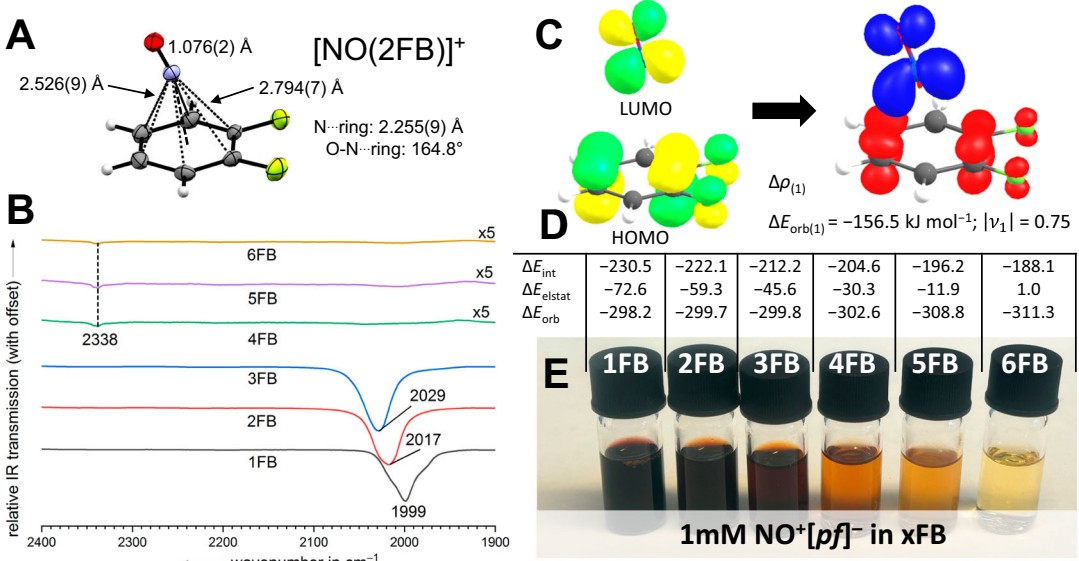

**Fig. 4 | Interaction of the nitrosonium cation with xFB solvents. A** Molecular structure of the cation in solid [NO(2FB)]⁺[*pf*]⁻, color code: fluorine−green, oxygen −red, nitrogen−blue, carbon−gray, hydrogen−light gray, anion omitted for clarity, thermal ellipsoids set at 50% probability. **B** ATR-IR spectra of the solid residues obtained from evaporated solutions of NO⁺[*pf*]⁻ in xFB (x = 1−6) between 1900 and 2400 cm⁻¹. The spectra from 4FB to 6FB are multiplied by the factor 5, due to the low intensity of the NO band of non-coordinated NO⁺ (calc. at: 28 km mol⁻¹) in comparison to the intensities of the NO bands in the Wheland complexes

(calc. at: >1000 km mol⁻¹). **C** The shape of the deformation density $\Delta\rho_{(1)}$ corresponding to $\Delta E_{orb(1)}$ (charge flow: red → blue) and the participating fragment orbitals of NO⁺ and 2FB at the BP86(D3BJ)/TZ2P level. The eigenvalue $|v_1|$ gives the amount of the charge migration in e⁻. **D** EDA-NOCV results for [(xFB)NO]⁺ complexes using NO⁺ and xFB as interacting fragments at the BP86(D3BJ)/TZ2P level given in kJ mol⁻¹. **E** Photo of the solutions of NO⁺[*pf*]⁻ in xFB (x = 1−6) at a concentration of 1 mM. Note the decreasing color intensity with increasing degree of fluorination of xFB.

interactions of NO⁺ with xFB in the solid state. For 1−3FB, the spectra were similar to the spectra in solution, with $\nu_{NO}$ increasing from 1999 (1FB) to 2029 cm⁻¹ (3FB). For residues stemming from 4 to 6FB solution, a weak $\nu_{NO}$ band was observed at 2338 cm⁻¹ (Fig. 4B) and hence essentially at the same position as in neat NO⁺[*pf*]⁻. In agreement with this, we observed the solvent-free crystallization of NO⁺[WCA]⁻ from 6FB (WCA = *pf*, *al-f-al*). Overall, we assume that the xFB⋯NO⁺ interaction for x = 4−6 is too weak to allow for co-crystallization as a Wheland complex.

**Molecular structures as a function of solvent properties.** Analysis of Figs. 3 and 4 gives some important clues on the species that exist in solution: Although $\varepsilon_s$(3FB) is the highest of all investigated solvents and one would therefore expect solvated ions to prevail (i.e., reduced ion-pairing), the scXRD structure corresponds to a tight ion-pair {(3FB)Ag[*pf*]}$_{ip}$ that saturates the Ag⁺ coordination sphere by contacts to the [*pf*]⁻ ion and one 3FB solvent molecule. And although $\varepsilon_s$ for 2FB and 4FB are virtually the same, only 2FB exhibits solvent-coordinated cations in the crystal structures [Ag(S)$_n$]⁺[*pf*]⁻ with n = 2 at room temperature and n = 3 for low-temperature crystallization. By contrast, 4FB forms a tight ion-pair {(4FB)Ag[*pf*]}$_{ip}$ similar to that found for 3FB. A similar consideration holds for the 1FB/5FB couple with comparable $\varepsilon_s$. Yet, the salt [Ag(1FB)$_3$]⁺[*pf*]⁻ forms in 1FB at all conditions tested, and from 5FB a tight ion-pair {Ag[*pf*]}$_{ip}$ without interaction to any solvent molecule is formed. Hence, it appears that with increasing fluorine content in xFB, the Ag⁺ ion complexation enthalpies $\Delta_rH_{complex}$ get sequentially weaker, so that starting with 3FB, the saturation of the Ag⁺ coordination sphere by the counterion [*pf*]⁻ is preferred over the solvent. Apparently, $\Delta_rH_{complex}$ for 5FB is so low that even the weak contacts of the tight ion-pair {Ag[*pf*]}$_{ip}$ to fluorine atoms of other [*pf*]⁻ ions in the lattice are more favorable than coordination of a 5FB solvent molecule. Interestingly, in CH₂Cl₂ with a $\varepsilon_s$ value between 1FB/5FB and 2FB/4FB, clear salt structures [Ag(S)$_n$]⁺[*pf*]⁻ with n = 3 at room temperature and even n = 4 at low temperature are formed. In 1,2-Cl₂C₂H₄, exclusively the salt type [Ag(S)₃]⁺[*pf*]⁻ with the more favorable 5-ring

chelates is formed. Therefore, the interaction of CH₂Cl₂ or 1,2-Cl₂C₂H₄ with the Ag⁺ ion must be at least similar in strength to 1FB, slightly stronger than Ag⁺ with 2FB, but much stronger than Ag⁺ with 3FB, 4FB and 5FB. This agrees with the formal potentials collected in Table 2.

**Quantum chemical calculations**

To quantify the extent of stabilization of the solvated species [de(S)$_n$]⁺ (n = 1−4) and to further evaluate the presence of ion-pairs {Ag[*pf*]}$_{ip,solv}$ and {(S)Ag[*pf*]}$_{ip,solv}$, the structures of all complexes [de(S)$_n$]⁺ (de = Ag, NO), {Ag[*pf*]}$_{ip,solv}$ and {(S)Ag[*pf*]}$_{ip,solv}$ were optimized at the (RI-)BP86(D3BJ)/def2-TZVPP[85] DFT[86] level of theory, refined and extrapolated to the complete basis set limit (CBS) in a series of DLPNO-CCSD(T) single point calculations[87−89]. Solvation was considered by the COSMO-RS[90−92] model and used to derive the gas-phase quantities $\Delta_rH°$(g) and $\Delta_rG°$(g) as well as $\Delta_rH°$(S) and $\Delta_rG°$(S) in S at standard conditions (g: 298 K, 1 bar; solv.: 298 K, a = 1 mol L⁻¹). Full data is presented in the Supplementary Notes 4.1−4.4.

All calculated gas-phase quantities $\Delta_rH°$(g) and $\Delta_rG°$(g), as well as $\Delta_rH°$(S) and $\Delta_rG°$(S) in S at standard conditions, are collected in full detail in Supplementary Note 4.3. In Fig. 5, we only discuss the solution-values $\Delta_rG°$(S) relevant for the development of the formal potential of Ag⁺(S)/Ag as a function of S. Overall, the consecutive and total solvent complexation Gibbs energies and enthalpies of Ag⁺ greatly decrease with increasing fluorination of xFB, both in S and in the gas phase. Typically, two or three xFB molecules may be taken up in the solvates [Ag(S)$_n$]⁺ (yellow, orange, red bars in Fig. 5). Yet, especially for the highly fluorinated xFB molecules with x = 4−5, the desolvation Gibbs energy $\Delta_rG°$(xFB) of 9 to 28 kJ mol⁻¹ is very low and, hence, the solvated Ag⁺ ion is very reactive in these solvents. This agrees with the very high $E°'$(Ag⁺(S)/Ag) values of 1.47 (S = 4FB) and 1.50 V (S = 5FB) vs. $E°'$(Fc⁺/Fc, S) measured in S. In addition, for the three most fluorinated xFB solvents, the calculations predict that ion-pairs are more favorable than dissociated ions (cf. yellow bars to green/blue bars in Fig. 5). Pleasingly, the complexes or ion-pairs, calculated to be most favorable in solution, comply well with those found with scXRD analyses (Fig. 3).

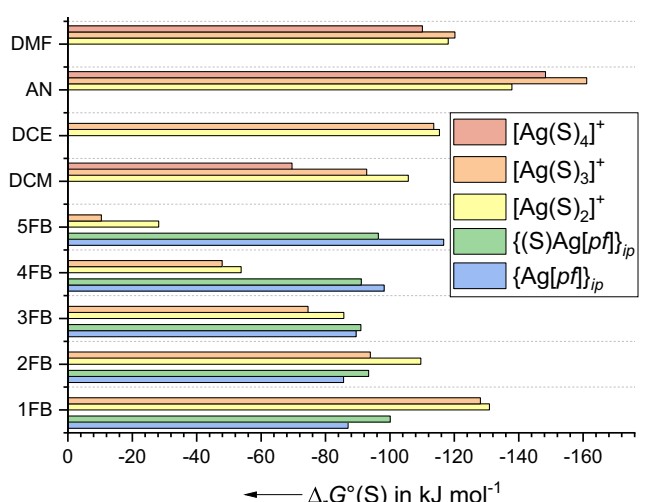

**Fig. 5 | Solvation Gibbs energies of the silver cation.** Calculated values of $\Delta_r G°(S)$ for the formation of the silver ion solvates ($Ag^+ + nS \rightarrow [Ag(S)_n]^+$) and ion-pairs ($Ag^+ + [pf]^- (+S) \rightarrow \{Ag[pf]\}_{ip}/\{(S)Ag[pf]\}_{ip}$) at the DLPNO-CCSD(T)/CBS level with COSMO-RS solvation in S. The individual values are reported in full detail in Supplementary Note 4.3. The species with the largest negative value of $\Delta_r G°(S)$ in the graph, is the most favorable one in solution in S at standard conditions. AN stands for $H_3CCN$, DCE for $1,2-Cl_2C_2H_4$ and DCM for $CH_2Cl_2$.

In addition, the complexation Gibbs energies of $Ag^+$ with 2FB and $CH_2Cl_2$ are comparable as are their $E°´(Ag^+(S)/Ag)$-values of +0.99 and +0.88 V vs. $E°´(Fc^+/Fc, S)$. Since $\Delta_r G°(S)$ calculated for 1FB is more favorable than for 2FB and $CH_2Cl_2$, the $E°´(Ag^+(1FB)/Ag)$ potential is reduced to 0.74 V. The potential difference between the $E°´(Ag^+(S)/Ag)$-values in 2FB and 1FB (0.99−0.74 = 0.25 V), also agrees with the difference of the sum of their 1st and 2nd complexation Gibbs energies in solution of 21.3 kJ mol$^{-1}$; this corresponds with $\Delta G = -zF\Delta E$ to 0.22 V (cf. Supplementary Note 4.3). For xFB solutions with x > 2, the values are influenced by ion-pairing and cannot be used for such quantitative evaluations. By contrast, the complexation Gibbs energies of AN, which induce the lowest $Ag^+$ potential of +0.02 V vs. $Fc^+/Fc$, are also the largest of all the solvents assessed within Fig. 5. This also complies with the experiment. Hence, our results suggest that the $E°´(Ag^+(S)/Ag)$ values collected in Table 2 reflect the true reactivity of the electroactive $Ag^+$ ions in S. They are induced by poor ligand capacities of xFB.

All calculated $NO^+$ interaction energies are collected in Supplementary Note 4.4. Interestingly, while the formation of Wheland complexes is favored for all xFB molecules in the gas phase and, in part, even the uptake of a second xFB molecule is viable, the situation is very different in xFB-solution: Apparently, only 1FB−with the lowest *IE* of all xFB molecules (Table 2)−is electron-rich enough to slightly favor the complex $[NO(1FB)]^+$ also in 1FB-solution.

By contrast, in all other cases, the calculations suggest the presence of uncomplexed $NO^+$ in xFB (x = 2−5) solution, although this is on the edge for 2FB, which complies with the determined scXRD structure $[NO(2FB)]^+[pf]^-$ (Fig. 4A). This strive toward the formation of free $NO^+$ presumably results from the rather high calculated solvation Gibbs energies for the isolated small $NO^+$ ion vs. the complexed $[NO(xFB)]^+$ systems. Overall, from 1FB to 5FB the interaction energies gradually get lower, concomitant with the increase of the solvent *IE*s. This aligns with the potentials $E_{1/2}$ collected in Table 2 that vary in xFB solution over +1.52 (4FB) − 1.11 (1FB) = 0.41 V. Also, the difference of $\Delta_r G°(xFB)$ for the first arene complexation between 4FB and 1FB amounts to 32.1−(−6.8) = 38.9 kJ mol$^{-1}$ or 0.40 V. Hence, the drastic increase of the $NO^+$ potentials results from the inferior interaction of the cation with the higher fluorinated xFB solvents and the presence of free $NO^+$ in the latter cases. In addition, the weak tendency to form Wheland complexes $[NO(xFB)]^+$ aligns with the *IE*s of the xFB solvents of

9.20−9.63 eV in Table 2: They are close to (1FB), or even higher (xFB, x = 2−5) than that of NO of 9.26 eV. This can explain why solutions of $NO^+[pf]^-$ may be stored for weeks in xFB solution without noticeable decomposition. Moreover, the visual inspection of these solutions indicates decreasing red coloration, the typical Wheland-complex color, with increasing fluorine content of xFB (Fig. 4E).

Additionally, we performed an energy decomposition analysis combined with the natural orbitals of chemical valence (EDA-NOCV). In line with the experimental observations and other DFT calculations, the total interaction energy ($\Delta E_{int}$) between the $NO^+$ fragment and the different xFB fragments gradually decreases with increasing degree of fluorination (Fig. 4D). Interestingly, the decrease of the total interaction energy is not of a decrease of the orbital interaction energies ($\Delta E_{orb}$), but rather the electrostatic interaction ($\Delta E_{elstat}$). The entire EDA-NOCV results, including analyses for benzene and mesitylene, are included in Supplementary Table 40.

## Extension of the positive potentials in xFB with innocent deelectronators

After having established that the combination of xFB solvents and $[pf]^-$ WCA can tune the potentials of the commonly used $NO^+$ and $Ag^+$ deelectronators to much higher values, we proceed to push these limits to even higher values by using different deelectronators. Compared to the positive solvent limits $E_{pos}$ collected in Table 2, the $Ag^+$ and $NO^+$ potentials do not stretch over the full $E_{pos}$ limits, which the xFB solvents can tolerate +1.82 to +2.67 V vs. $E°´(Fc^+/Fc, S)$. Moreover, we have noted numerous times in synthetic applications that both, $Ag^+$ and $NO^+$ deelectronator reagents react *non-innocent* with substrates in xFB solution −with complexation[33,35,93–95], substitution[30,32,34,37] or degradation[33] being the key side−or even major reactions complicating or hindering the desired reaction outcome[64]. In addition, also the *magic blue*-like salts $[TBPA]^{+}[WCA]^-$, praised[5] for their innocent behavior and even if partnered with the excellent counterion $[pf]^-$, yield after deelectronations a neutral amine, that often does react *non-innocent*[41].

**Innocent deelectronators.** To generate a desired deelectronated target system $[M]^+(S)$ (Fig. 6, Eq. 1a), the secondary system in Eq. 1b needs to have a suitably high deelectronation potential $E°´(iD^+/iN, S)$, to be non-reactive toward both, the solvent S and the generated reactive cation $[M]^+$, and must be compatible with the counterion[17], i.e., the very good WCAs $[pf]^-$ and $[al\text{-}f\text{-}al]^{-}$[25,26]. Toward this goal, a series of innocent redox-couples $iD^+/iN$ was developed, from which we already published phenazine[F41] and anthracence[Hal33]. Structural formulae of the neutral iN are shown in Fig. 6, together with their deelectronation potentials $E°´(iD^+/iN, S)$ vs. $E°´(Fc^+/Fc, S)$ in 2FB or 4FB solution. The $iD^+/iN$-couples ($iN$ = anthracene$^F$, phenantrene$^F$) in Fig. 6 extend the positive deelectronation range in 4FB up to +1.89 V and considerably higher than the maximum of +1.52 V available with $E_{1/2}(NO^+, 4FB)$ from Table 2. In addition, all deelectronated $iD^+$ radical cations from Fig. 6 may be prepared as room temperature stable $iD^+[WCA]^-$ salts in 74 to 78% yield (WCA = $al\text{-}f\text{-}al$). For the systems, also the CV traces at scan rates between 20 and 1000 mV s$^{-1}$ and the scXRD structures of $iD^+[WCA]^-$ are included in Fig. 6 and Supplementary Notes 3.4.3−3.4.4. Note that for phenazine$^F$ and anthracene$^X$ (X = Hal, F) salts, their easiest preparation now follows the deelectronation of their iN neutrals in 4FB with the readily available $NO^+[WCA]^-$ salts[26,36], since their formal potentials $E°´(iD^+/iN, 4FB)$ of +1.29−1.47 V are below the +1.52 V potential of $NO^+[WCA]^-$ in 4FB. In addition, we have shown that the silver salt $Ag^+[pf]^-$ dissolved in 4FB and with a formal potential of +1.47 V is suited to quickly deelectronate anthracene$^{Hal}$ ($E°´ = 1.42$ V) and slower also anthracene$^F$ ($E°´ = 1.47$ V) with formation of solid silver $Ag^0(s)$ in the reaction (Supplementary Notes 3.8). However, to ionize the phenantrene$^F$ neutral, one needs to use the silver salts $Ag^+[WCA]^-$ in combination with 0.5 equivalents of $I_2$. The concomitantly formed solid AgI provides the additional thermodynamic driving force to deelectronate the neutral iN. Note, that the

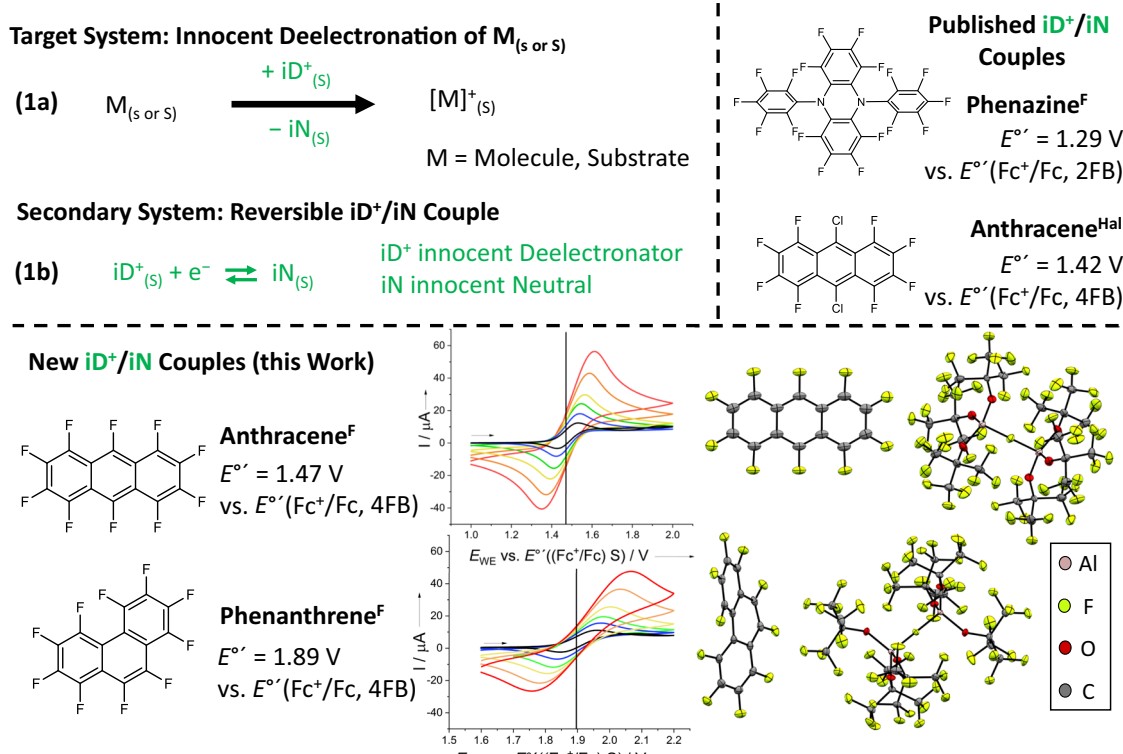

**Fig. 6 | Key data of the innocent deelectronators.** Top: Requirements for innocent deelectronation described by Eq. 1a–b and structural formulae and formal potentials $E°'(iD^+, S)$ of the known[33,41] innocent deelectronator/neutral couples. Bottom: Structural formulae and formal potentials $E°'(iD^+, S)$ of the innocent deelectronator/neutral couples with increased potential to be used for innocent deelectronation reactions. Middle: Chemically reversible electronation and deelectronation in cyclic voltammograms (2nd cycle) of anthracene[F] and phenantrene[F] at different scan rates (black to red, 20–1000 mV s$^{-1}$) of phenanthrene[F] (10 mM) in 4FB using [NBu$_4$]$^+$[Al(OR$^f$)$_4$]$^-$ (100 mM) as supporting electrolyte. Right: Molecular structures of deelectronated anthracene[F] and phenantrene[F] as [al-f-al]$^-$ salts. Thermal ellipsoids set at 50% probability.

Ag$^+$[WCA]$^-$/0.5 I$_2$ system is only fully compatible with 4FB and higher fluorinated benzenes. We have estimated the limit for deelectronation with Ag$^+$[WCA]$^-$/0.5 I$_2$ to roughly amount to +2.3–2.4 V vs. $E°'$(Fc$^+$/Fc, S), since $E_{pos}$(3FB) = +2.35 V (Table 2) and 3FB already reacts with the synergistic mixture Ag$^+$[WCA]$^-$/0.5 I$_2$.

Favorable for any application, all iD$^+$[WCA]$^-$ salts—prepared from the iN molecules in Fig. 6—are long-term stable at room temperature (at least several months), given inert and anhydrous conditions. Hence, we expect innocent deelectronation with iD$^+$[WCA]$^-$ salts in inert xFB solvents to largely extend the limits of currently possible deelectronation chemistry. All of this works using standard techniques and glassware with commercially available reagents. Hence, from this combination of reagents and solvents a wide variety of cationic chemical systems may be synthesized that potentially stretch from organic carbocations, over transition metal carbonyl and other organometallic cations, to even non-metal cations. Since the four iN in Fig. 6 are also reversibly addressable at an inert electrode, the pairs xFB / iN also do hold large promise as redox mediators, e.g., for electrosynthesis and -catalysis or maybe also as redox shuttle to mediate cut-off potentials in batteries[96–101].

## Discussion

Here we present the very favorable properties of the strategically and dipole-maximized substituted fluorobenzenes xFB as inert solvents that are tolerant to high deelectronation potentials. With increasing fluorination and if paired with the good WCAs [pf]$^-$ or [al-f-al]$^-$, the xFB molecules turn into increasingly poor ligands and induce markedly increasing deelectronation potentials of the classical deelectronator ions Ag$^+$ and NO$^+$ that may reach values as high as +1.50/+1.52 V vs. $E°'$(Fc$^+$/Fc, S). Concomitantly, this combination enables very good

solubilities of the de$^+$[WCA]$^-$ salts with high activities of the solvated de$^+$ ions. The most relevant system properties for potential applications in many fields of chemistry or the material sciences are compiled in Table 3.

**Table 3 | Properties relevant to apply the combination xFB with deelectronators de[WCA] and [NBu$_4$]$^+$[pf]$^-$ as supporting electrolyte salt**

| Fluorobenzenes S→ ↓ Properties | 1FB | 2FB | 3FB | 4FB | 5FB |
|---|---|---|---|---|---|
| Melting point/°C | −42 | −47 | −15 | −40 | −47 |
| Boiling point/°C | +84 | +94 | +95 | +94 | +85 |
| $E_{neg}$ vs. $E°'$(Fc$^+$/Fc, S)$^a$/V | −3.18 | −3.09 | −3.02 | −3.08 | −2.37 |
| $E_{pos}$ vs. $E°'$(Fc$^+$/Fc, S)$^a$/V | +1.82 | +2.05 | +2.35 | +2.43 | +2.67 |
| $E_{1/2}$(NO$^+$(S)/NO)$^b$/V | +1.11 | +1.23 | +1.42 | +1.52 | +1.47 |
| $E°'$(Ag$^+$(S)/Ag)$^b$/V | +0.74 | +0.99 | +1.26 | +1.47 | +1.50 |
| Ionization Energy IE$^c$/eV | 9.20 | 9.29 | 9.40 | 9.53 | 9.63 |
| Static dielectric constant$^d$ ($\varepsilon_r$) | 5.8 | 14.2 | 22.1 | 12.6 | 4.5 |
| Dipole moment$^{d,e}$/D | 1.9 | 3.5 | 4.6 | 3.5 | 1.8 |
| Dynamic viscosity$^d$ $\eta$/mPa•s | 0.5662 | 0.6217 | 0.6962 | 0.7826 | 0.8165 |
| Density$^d$ [$T$]/g cm$^{-3}$ [°C] | 1.022 [23] | 1.156 [23] | 1.292 [23] | 1.401 [23] | 1.511 [24] |

$^a$Using 100 mM [NBu$_4$]$^+$[pf]$^-$ as supporting electrolyte salt in xFB.
$^b$vs. $E°'$(Fc$^+$/Fc, S).
$^c$CV measurements of 10 mM solutions using 100 mM [NBu$_4$]$^+$[pf]$^-$ as supporting electrolyte salt in xFB at a scan rate of 100 mV s$^{-1}$.
$^d$Measurements at (22.0 ± 0.5) °C for $\varepsilon_r$, D and $\eta$, at $T$ for $\rho$.
$^e$Taken from relaxation strengths of the liquid phase—typically higher than gas phase due to induced dipoles.

If these de$^+$[WCA]$^-$ reagents (de = Ag, NO) do react non-innocently by coordination, substitution or degradation and despite the solvents being inert, one can switch to the room temperature stable innocent deelectronator salts iD$^+$[WCA]$^-$ with [anthracene$^F$]$^{+\bullet}$ and [phenantrene$^F$]$^{+\bullet}$ radical cations that further push the limits of reversibly addressable deelectronation chemistry up to +1.89 V vs. $E°´$(Fc$^+$/Fc, 4FB). Since these potentials are also available at an inert electrode, we suggest that these combinations of innocent solvent and innocent deelectronator may also be advantageously used as redox mediators for electrosynthesis and catalysis or as redox shuttles in battery electrolytes. Favorably, the innocent solvent properties may be fine-tuned by mixing. For example, at room temperature static dielectric constants $\varepsilon_r$ between 2 and 22 may be realized for binary xFB systems.

## Methods

### Syntheses

The used syntheses for the nitrosyl, silver and [NBu$_4$]$^+$ salt were already described in earlier publications of our group[36,102,103]. Nonetheless, it is worth mentioning again that the synthesis for the nitrosyl salt could be done in a at least 10 gram scale with regard to the lithium salt, using the commercial NO$^+$[BF$_4$]$^-$ salt and without lithium impurities in the product[36,104]. The synthesis of the solvent-free silver salt was a great progress to avoid potential harmful dichloromethane in further reactions[25,29,102]. Additionally, we could develop a new synthetic route for the known Fc$^+$ salt using the nitrosyl, instead of the silver salt as oxidizing agent, to avoid colloidal silver which could be disturbing during further experiments. For a better allocation in the following NMR spectra, instead of the abbreviation [pf]$^-$ the detailed description [Al{OC(CF$_3$)$_3$}$_4$]$^-$ will be used.

### Nitrosonium tetrakis(nonafluorotertbutanolato)aluminate(III) (NO$^+$[pf]$^-$)

**Caution!** This procedure involves the work with liquified sulfur dioxide, which has a vapor pressure of ca. 4 bar at room temperature. Therefore, the synthesis requires trained personnel and proper equipment.

NO$^+$[Al{OC(CF$_3$)$_3$}$_4$]$^-$ was synthesized based on the procedure published in ref. 36: Li$^+$[pf]$^-$ (9.74 g, 10 mmol, 1.0.eq) and NO$^+$[BF$_4$]$^-$ (1.75 g, 15 mmol, 1.5 eq.) were filled in one side of a double-bulb Schlenk vessel equipped with a G4 frit plate. Sulfur dioxide (10 mL) was condensed onto the mixture of the reagents at −78 °C. The vessel was brought to room temperature and stirred for 7 days. Afterward, the solution was filtered and the solvent was removed by vacuum. NO$^+$[pf]$^-$ was obtained as a colorless solid (8.86 g, 89%).

Characterization

$^1$**H-NMR** (300.18 MHz, CD$_2$Cl$_2$, calibration to CHDCl$_2$ = 5.32 ppm[105], 298 K): No signals observed.

$^7$**Li-NMR** (116.66 MHz, CD$_2$Cl$_2$, 298 K): No signals observed.

$^{11}$**B-NMR** (96.31 MHz, CD$_2$Cl$_2$, 298 K): No signals observed.

$^{14}$**N-NMR** (21.69 MHz, CD$_2$Cl$_2$, 298 K): δ = 364.5 (s, **NO$^+$**, 1 N) ppm.

$^{19}$**F-NMR** (282.45 MHz, CD$_2$Cl$_2$, 298 K): δ = −75.7 (s, [Al{OC(**CF$_3$**)$_3$}$_4$]$^-$, 36 F) ppm.

$^{27}$**Al-NMR** (78.22 MHz, CD$_2$Cl$_2$, 298 K): δ = 34.7 (s, [**Al**{OC(CF$_3$)$_3$}$_4$]$^-$, 1Al) ppm.

**FTIR** (ZnSe, ATR):ν/cm$^{-1}$ = 2342 (vw), 1354 (vw), 1301 (m), 1248 (vs), 1209 (vs), 968 (vs), 863 (vw), 830 (vw), 757 (vw), 726 (vs), 642 (vw), 568 (vw), 560 (vw).

**FT Raman** (1000 scans, 250 m0W):ν/cm$^{-1}$ = 2937 (vw), 2756 (vw), 2340 (m), 1355 (vw), 1315 (vw), 1275 (w), 1248 (vw), 1204 (vw), 1120 (vw), 975 (vw), 829 (vw), 815 (vw), 796 (vs), 746 (s), 571 (w), 562 (w), 537 (m), 366 (m), 325 (vs), 290 (m), 237 (m), 208 (w), 174 (w), 118 (m), 100 (m).

### Silver tetrakis(nonafluorotertbutanolato)aluminate(III) (Ag$^+$[pf]$^-$)

**Caution!** This procedure involves the work with liquified sulfur dioxide, which has a vapor pressure of ca. 4 bar at room temperature. Therefore, the synthesis requires trained personnel and proper equipment.

Ag$^+$[Al{OC(CF$_3$)$_3$}$_4$]$^-$ was synthesized based on the procedure published in ref. 102 The reaction was performed analogously to the one yielding NO$^+$[pf]$^-$. Instead of NO$^+$[BF$_4$]$^-$, AgF (1.91 g, 15 mmol, 1.5 eq.) was used. Additionally, the reaction was performed in the absence of light. Ag$^+$[pf]$^-$ was obtained as a colorless solid (9.03 g, 84%).

Characterization

$^1$**H-NMR** (400.17 MHz, CD$_2$Cl$_2$/Et$_2$O, calibration to CHDCl$_2$ = 5.32 ppm[105], 298 K): 3.50 (q, $^3J_{HH}$ = 7.0 Hz, O(**CH$_2$**CH$_3$)$_2$, 2H) and 1.20 (t, $^3J_{HH}$ = 7.0 Hz, O(CH$_2$**CH$_3$**)$_2$, 3H) ppm.

$^{19}$**F-NMR** (282.45 MHz, CD$_2$Cl$_2$/Et$_2$O, 298 K): δ = −75.8 (s, [Al{OC(**CF$_3$**)$_3$}$_4$]$^-$, 36 F) ppm.

$^{27}$**Al-NMR** (104.27 MHz, CD$_2$Cl$_2$/Et$_2$O, 298 K): δ = 34.6 (s, [**Al**{OC(CF$_3$)$_3$}$_4$]$^-$, 1Al) ppm.

$^7$**Li-NMR** (116.7 MHz, CD$_2$Cl$_2$/Et$_2$O $_2$, 298 K): No signals observed.

### Tetrabutylammonium tetrakis(nonafluorotertbutanolato)aluminate(III) ([NBu$_4$]$^+$[pf]$^-$)

[NBu$_4$]$^+$[Al{OC(CF$_3$)$_3$}$_4$]$^-$ was synthesized based on the procedure published in ref. 103: Li$^+$[pf]$^-$ (19.5 g, 20 mmol, 1.0 eq.) and [NBu$_4$]$^+$Br$^-$ (6.45 g, 20 mmol, 1.0 eq.) were dissolved in a mixture of water and acetone (85:15 v/v, 150 mL) at room temperature. The solution was kept at a warm place/heated at 30 °C overnight, allowing the acetone in the solvent to evaporate, yielding a microcrystalline precipitate. The remaining solvent was removed by filtration and the residue was washed with water until all the bromide was removed (test with, e.g., silver nitrate). Afterward, the product was washed two times with hexane (2 × 100 mL). [NBu$_4$]$^+$[pf]$^-$ was obtained as a colorless powder (22.9 g, 94%).

Characterization

$^1$**H-NMR** (300.18 MHz, CD$_2$Cl$_2$, calibration to CHDCl$_2$ = 5.32 ppm[105], 298 K): δ = 3.07 (m, [N(**CH$_2$**CH$_2$CH$_2$CH$_3$)$_4$]$^+$, 8H), 1.60 (m, [N(CH$_2$**CH$_2$**CH$_2$CH$_3$)$_4$]$^+$, 8H), 1.43 (m, [N(CH$_2$CH$_2$**CH$_2$**CH$_3$)$_4$]$^+$, 8H), 1.03 (t, $^3J_{HH}$ = 7.3 Hz [N(CH$_2$CH$_2$CH$_2$**CH$_3$**)$_4$]$^+$, 12H) ppm.

$^{19}$**F-NMR** (282.45 MHz, CD$_2$Cl$_2$, 298 K): δ = −75.7 (s, [Al{OC(**CF$_3$**)$_3$}$_4$]$^-$, 36 F) ppm.

$^{27}$**Al-NMR** (78.22 MHz, CD$_2$Cl$_2$, 298 K): δ = 34.6 (s, [**Al**{OC(CF$_3$)$_3$}$_4$]$^-$, 1Al) ppm.

$^7$**Li-NMR** (116.7 MHz, CD$_2$Cl$_2$, 298 K): No signals observed.

$^{14}$**N-NMR** (21.9 MHz, CD$_2$Cl$_2$, 298 K): No signals observed.

### Bis($\eta^5$-cyclopentadienyl)iron(III) tetrakis(nonafluorotertbutanolato)aluminate(III) (Fc$^+$[pf]$^-$)

NO$^+$[pf]$^-$ (1.00 g, 1.01 mmol, 1.00 eq.) and Fc (0.23 g, 1.23 mmol, 1.22 eq.) were weighed, inside a glovebox, in one side of a double-Schlenk tube separated by a G3 or G4 frit and equipped with grease-free PTFE valves. Under reverse flow of Argon, 2FB (1,2-difluorobenzene, 10 mL) was added and led immediately to the formation of NO$_{(g)}$ and a dark blue solution. The solution was stirred at RT overnight and the solvent was removed under vacuo. To remove the excess of ferrocene, the residue was washed with n-hexane (5 mL). Therefore, the ferrocene solution in n-hexane was filtered through the frit and the solvent was condensed back to the side of the crude product, as many times, as the n-hexane solution was still colored yellowish before the filtration. Afterward, the crude product was dried under vacuo (10$^{-3}$ mbar) to yield a blue powder of Fc$^+$[Al{OC(CF$_3$)$_3$}$_4$]$^-$ (0.99 g, 0.86 mmol, 85%).

Characterization

$^1$**H-NMR** (300.18 MHz, 1,2-F$_2$C$_6$H$_4$ (2FB), calibration to 1,2-F$_2$C$_6$H$_4$ = 6.96 ppm against Si(CH$_3$)$_4$, 298 K): 33.87 (br. s., [Fe(C$_5$H$_5$)$_2$]$^+$, 10H) ppm.

**$^{19}$F-NMR** (282.45 MHz, 2FB, 298 K): δ = −75.7 (s, [Al{OC(C**F**$_3$)$_3$}$_4$]$^−$, 36 F), −139.6 (s, 1,2-**F**$_2$C$_6$H$_4$, 2 F) ppm.

**$^{27}$Al-NMR** (78.22 MHz, 2FB, 298 K): δ = 34.7 (s, [**Al**{OC(CF$_3$)$_3$}$_4$]$^−$, 1Al) ppm.

**FTIR** (ZnSe, ATR):ν/cm$^{-1}$ = 3126 (vw), 1423 (vw), 1352 (vw), 1299 (w), 1273 (m), 1266 (m), 1253 (m), 1239 (m), 1213 (vs), 1163 (w), 1064 (vw), 1014 (vw), 972 (vs), 856 (w), 832 (vw), 792 (vw), 756 (vw), 728 (vs), 571 (vw).

**FT Raman** (1000 scans, 200 mW):ν/cm$^{-1}$ = 3133 (vw), 1425 (vw), 1363 (vw), 1304 (vw), 1273 (vw), 1113 (m), 1065 (vw), 851 (vw), 746 (vw), 562 (vw), 538 (vw), 367 (vw), 321 (w), 299 (vs), 234 (vw), 170 (vw), 120 (vw), 82 (vw).

### Nitrosonium bis{tris(nonafluorotertbutanolato)aluminum(III)}-($μ_2$)-fluoride (NO$^+$[al-f-al]$^−$)

**Caution!** This procedure involves the work with liquified sulfur dioxide, which has a vapor pressure of ca. 4 bar at room temperature. Therefore, the synthesis requires trained personnel and proper equipment.

NO$^+$[F{Al(OC(CF$_3$)$_3$}$_2$]$^−$ was synthesized based on the procedure published in ref. 26. NO$^+$[PF$_6$]$^−$ (560 mg, 3.20 mmol, 1.0 eq.) and (H$_3$C)$_3$Si−F−Al(OC(CF$_3$)$_3$)$_3$ (5.04 g, 6.1 mmol, 2.0 eq.) were filled in a Schlenk vessel inside a glovebox. Sulfur dioxide (10 mL) was condensed onto the mixture of the reagents at −78 °C. The vessel was equipped with a bubbler, brought to −35 °C and the temperature was held for 1 h. Subsequently, the reaction solution was slowly warmed to room temperature and the sulfur dioxide evaporated. The white powder was dried at 10$^{-3}$ mbar for 2 h. [NO]$^+$[al-f-al]$^−$ was obtained as a colorless powder (4.36 g, 90%).

### Silver bis{tris(nonafluorotertbutanolato)aluminum(III)}-($μ_2$)-fluoride (Ag$^+$[al-f-al]$^−$)

**Caution!** This procedure involves the work with liquified sulfur dioxide, which has a vapor pressure of ca. 4 bar at room temperature. Therefore, the synthesis requires trained personnel and proper equipment.

Ag$^+$[F{Al(OC(CF$_3$)$_3$}$_2$]$^−$ was synthesized based on the procedure published in ref. 26. The reaction was performed analogously to the one yielding NO$^+$[al-f-al]$^−$. Instead of NO$^+$[PF$_6$]$^−$, Ag$^+$[PF$_6$]$^−$ (810 mg, 3.20 mmol, 1.0 eq.) was used. Additionally, the reaction was performed in the absence of light. Ag$^+$[al-f-al]$^−$ was obtained as a colorless solid (4.68 g, 92%).

### Decafluoroanthracene (anthracene$^F$)

Anthracene$^F$ was synthesized based on the procedure published in ref. 106: 9,10-dichlorooctafluoroanthracene (2 g, 5.11 mmol, 1.0 eq.) and KF (0.98 g, 16.9 mmol, 3.3 eq.) were dissolved in a mixture of sulfolane (10 mL) and toluene (20 mL). The reaction mixture was heated for 2 h at 120 °C. Afterward, the toluene was removed from the reaction mixture. The reaction mixture was further heated for 4 h at 210 °C. At room temperature, water (50 mL) was added to the mixture. The mixture was filtrated and the brown residue was washed with water (3 × 20 mL). After drying at room temperature, the residue was taken up in dichloromethane (20 mL) and slowly cooled down to −40 °C. The solution was decanted and anthracene$^F$ was obtained as a yellow to brown solid (0.63 g, 32%).

### Decafluoroanthracenium bis{tris(nonafluorotertbutanolato) aluminum(III)}-($μ_2$)-fluoride ([anthracene$^F$]$^+$[al-f-al]$^−$)

[NO]$^+$[F{Al(OC(CF$_3$)$_3$}$_2$]$^−$ (0.10 g, 1.0 eq, 66 μmol) and anthracene$^F$ (28 mg, 1.2 eq., 78 μmol) were placed in a Schlenk flask and 1,2,3,4-tetrafluorobenzene (1 mL) was added. The solution instantaneously turned green-blue and a gas formation was observed. The solution was stirred 5 min at room temperature and was then layered with n-pentane (10 mL). Slow diffusion of the solvents over days led to the crystallization of [anthracene$^F$]$^+$[F{Al(OC(CF$_3$)$_3$}$_2$]$^−$ in blue plates, suitable for scXRD (90 mg, 49 μmol, 74%).

#### Characterization

**FTIR** (ZnSe, ATR):ν/cm$^{-1}$ = 1600 (vw), 1517 (vw), 1491 (w), 1468 (vw), 1443 (vw), 1355 (w), 1300 (w), 1265 (s), 1243 (vs), 1211 (vs), 1177 (s), 1118 (w), 1038 (vw), 970 (vs), 957 (s), 863 (w), 810 (vw), 760 (vw), 726 (vs), 716 (m), 664 (w), 631 (w), 569 (w).

**FT-Raman** (1000 scans, 50 mW):ν/cm$^{-1}$ = 1570 (s), 1548 (s), 1436 (vs), 1415 (s), 1395 (vs), 1294 (s).

### Decafluorophenanthrenium bis{tris(nonafluorotertbutanolato) aluminum(III)}-($μ_2$)-fluoride ([phenanthrene$^F$]$^+$[al-f-al]$^−$)

Ag$^+$[F{Al(OC(CF$_3$)$_3$}$_2$]$^−$ (84 mg, 1.0 eq., 53 μmol), I$_2$ (6.7 mg, 0.5 eq., 27 μmol) and phenanthrene$^F$ (19 mg, 1.0 eq., 53 μmol) were placed in a Schlenk flask and 1,2,3,4-tetrafluorobenzene (0.8 mL) was added to the solution. The solution was filtered and brownish crystals suitable for scXRD formed out of the dark solution upon slow removal of the solvent. The product was obtained as dark brown crystalline blocks (78 mg, 42 μmol, 78%).

#### Characterization

**FTIR** (ZnSe, ATR):ν/cm$^{-1}$ = 1673 (vw), 1662 (vw), 1645 (vw), 1584 (vw), 1522 (w), 1495 (m), 1468 (vw), 1433 (vw), 1379 (w), 1354 (vw), 1301 (w), 1265 (m), 1242 (vs), 1213 (vs), 1180 (m), 1105 (vw), 1093 (w), 1069 (vw), 1011 (vw), 972 (vs), 866 (vw), 847 (m), 799 (vw), 727 (vs), 707 (m), 636 (w), 569 (w).

**Dielectric spectroscopy.** Complex permittivity spectra were recorded using an Anritsu MS4647A vector network analyzer connected to an open-ended coaxial probe at frequencies ranging from 1 to 70 GHz[107]. The reflectometer was calibrated using air, conductive silver paint, and N,N-dimethylacetamide[108]. Samples were placed into a double-walled sample holder connected to a Julabo-E12 thermostat to control the temperature at 5, 10, 18, 25, 33, or 40 °C. All complex permittivity spectra were modeled using a Debye relaxation[70] to obtain $ε_s$, the low-frequency limit of the permittivity, $ε_∞$, the high-frequency limit of the permittivity, and the dielectric relaxation time $τ$. Recorded spectra, together with the details of the analysis, are given in Supplementary Note 5.

**Cyclovoltammetry, general procedures.** All cyclic voltammograms were recorded in an argon-filled glovebox. A three-electrode arrangement was used with a 1 mm Pt disc working electrode (WE), a Pt mesh or wire as counter electrode (CE) and a Pt wire as pseudo reference electrode (RE) for NO$^+$ (Supplementary Notes 3.1), ferrocene (Fc) and N(4-BrC$_6$H$_4$)$_3$ or Ag wire for the Ag$^+$ and NO$^+$ evaluation (triangular) measurements in the Supplementary Notes 3.5.3 / 3.7.3. For all measurements [NBu$_4$]$^+$[pf]$^−$ (c = 100 mM) was used as supporting electrolyte. For each measured electroactive sample, c(de$^+$) = 10 mM was used. The RE (Pt|Fc$^+$, Fc or Ag$^+$|Ag reference) was added in a glass compartment with a frit to allow for a direct measurement and was filled with solutions of [NBu$_4$]$^+$[pf]$^−$ (100 mM), Fc (10 mM) and Fc$^+$[pf]$^−$ (10 mM) or [NBu$_4$]$^+$[pf]$^−$ (100 mM) and Ag$^+$[pf]$^−$ (10 mM). For each solvent S, a potential stability window (ECW) was recorded in a solution of [NBu$_4$]$^+$[pf]$^−$ (100 mM) in S to identify any impurities and to determine the stability range of the pure solvent against the RE used. Scan rates were varied from 20 mV s$^{-1}$, 50 mV s$^{-1}$, 100 mV s$^{-1}$ up to 200 mV s$^{-1}$, and if not stated otherwise, the half-wave potentials $E_{1/2}$ did not change with the rate (full details in Supplementary Note 3). Since several of the published potentials in the Geiger/Connelly Review[5] may have been afflicted by ion-pairing and other effects, we also measured the Ag$^+$ and NO$^+$ potentials in the like setup, but the solvents CH$_2$Cl$_2$, 1,2-Cl$_2$C$_2$H$_4$, dimethylformamide (DMF), acetonitrile (AN). In addition, the ECWs of nitromethane (MeNO$_2$), propylene carbonate (PC) and tetrahydrofurane (THF) were determined with [NBu$_4$]$^+$[pf]$^−$ (100 mM) as supporting electrolyte salt for comparison.

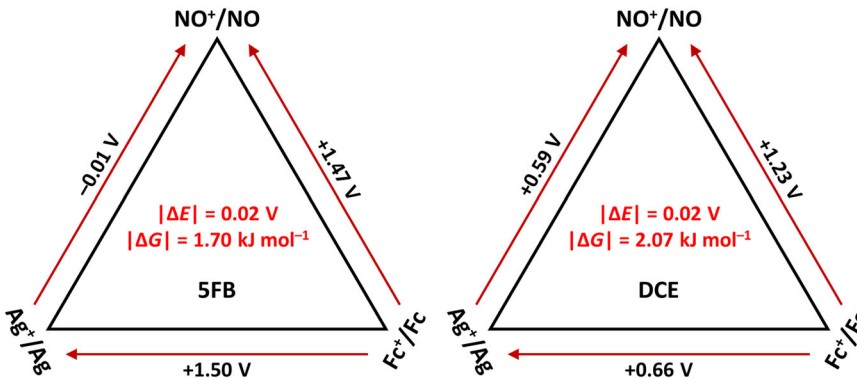

**Fig. 7 | CV-evaluation by triangular Born-Fajans-Haber-Cycles.** Measured half-wave potentials of [Fc]⁺ and [NO]⁺ versus Ag⁺(10 mₘ)/Ag and [NO]⁺ versus [Fc]⁺/Fc in exemplarily selected 5FB and DCE = 1,2-Cl₂C₂H₄ solution at a scan rate of 100 mV s⁻¹. Knowing two out of the three values of the measurements, the third can be calculated in a Born-Fajans-Haber-Cycle approach. Hence, |ΔE| and |ΔG| errors can be calculated by using the relation $\Delta G = -zF\Delta E$, with $z$ = number of electrons, $\Delta E$ = potential difference, $F$ = Faraday constant = 96,485 C mol⁻¹.[68] The mean errors of the calculated potential difference (ΔE) / the corresponding Gibbs energy difference (ΔG) are given in red in the center of the triangle. Note, $E_{1/2}$ potentials were rounded to two decimal places and rounding errors can occur.

**CV-evaluation by triangular Born-Fajans-Haber-Cycles.** This evaluation is exemplarily shown for the two solvents 5FB and 1,2-Cl₂C₂H₄ in Fig. 7, all other triangular cycles are deposited in the Supplementary Notes 3.5.3 / 3.7.3.

**Quantum chemical calculations.** An extended search of the potential energy surface was manually performed to find the lowest energy structures of all particles [de(S)ₙ]⁺ (de = Ag, NO), {Ag[pf]}ᵢₚ,ₛₒₗᵥ and {(S)Ag[pf]}ᵢₚ,ₛₒₗᵥ at the dispersion[109,110] corrected (RI-)BP86(D3BJ)/def2-TZVPP[85] DFT[86] level of theory. Corrections to statistical thermodynamics (ZPE, $H°$ and $S°$) were taken from the frequency calculations[111] at this level. The energies of the DFT structures were refined in a series of DLPNO-CCSD(T) single point calculations[87–89] with Dunning's[112–117] basis sets cc-pVDZ, cc-pVTZ, cc-pVQZ and then extrapolated to the complete basis set limit (CBS). These accurate CCSD(T)/CBS values were used as electronic energies for the calculation of the thermodynamics of all particles, augmented by DFT-corrections to ZPE, $H°$ and $S°$. Finally, contributions of solvation enthalpies and free energies in S were calculated with the COSMO-RS[90–92] model at the BP86(D3)/def2-TZVPD//BP86(D3)/def-TZVP level, so that we overall derive the quantities $\Delta_r H°(g)$ / $\Delta_r G°(g)$ in the gas phase as well as $\Delta_r H°(solv)$ / $\Delta_r G°(solv)$ in solution in S at standard conditions (g: 298 K, 1 bar; solv.: 298 K, $a = 1$ mol L⁻¹).

## Data availability

The X-ray crystallographic coordinates for structures reported in this study have been deposited at the Cambridge Crystallographic Data Centre (CCDC), under deposition numbers 2303021–2303026 and 2177466. These data can be obtained free of charge from The Cambridge Crystallographic Data Centre via www.ccdc.cam.ac.uk/data_request/cif. Processed data from all methods are included in the Supplementary Information. The raw data of all results presented in the manuscript or in the Supplementary Information are available from the corresponding authors (J.H. for DCS, I.K. for all other data). Additionally, the coordinates of computationally optimized structures are available in a text file in the Source Data Section. Source data are provided with this paper.

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

## Acknowledgements

This work was supported by the Albert-Ludwigs-Universität Freiburg and by the Deutsche Forschungsgemeinschaft (DFG, German Research Foundation)—Project numbers 431116391 (I.K.), 281091989 (I.K.), 350173756 (I.K.) and the European Research Council—Grant agreement ID: 101052935 (I.K.). M.Sellin is grateful for a PhD fellowship from the "Fonds der Chemischen Industrie". We would like to thank Fadime Bitgül and Dr. Harald Scherer for the measurement of the NMR spectra, Martina Knecht for performing the dielectric spectroscopy experiments, Dr. Manuel Schmitt, and Regina Stroh for valuable discussions of the DFT and ab initio calculations, Andreas Warmbold for the DSC measurement, Dr. Burkhard Butschke for his scXRD work. The authors acknowledge support by the state of Baden-Württemberg through bwHPC and the German Research Foundation (DFG) through grant no INST 40/467-1 and 40/575-1 FUGG (JUSTUS1 and 2 cluster).

## Author contributions

C.A. performed most of the CV measurements (with support from F.O., T.S.), the ab initio calculations and the measurements for the kinematic/dynamic viscosities of all xFB solvents, M.Sellin the syntheses and all characterizations/analyses of the innocent deelectronators and experiments concerning the nitrosonium salts together with M.Seiler and J.F. M.Sellin refined the solid-state structures. C.A., M.Sellin and I.K. wrote the manuscript. V.R. and M.Schmucker supervised the electrochemical measurements and T.W. measured the PFSGE-NMR-diffusion constants of ferrocene and cobaltocenium. J.H. performed the dielectric spectroscopy measurements and wrote the respective sections. I.K. supervised and conceptionally devised the project.

## Funding

## Competing interests
