## [Peer Review File · Nature Communications]

Pushing redox potentials to highly positive values using inert fluorobenzenes and weakly coordinating anionsREVIEWER COMMENTS

Reviewer #1 (Remarks to the Author):

This is an interesting paper that presents the results of a substantial body of experimental work on the series of fluorinated benzenes as solvents, particularly with regard to their application as weakly coordinating solvents for use with weakly coordinating anions.

The work is probably publishable but not until it has been completely rewritten to correct the serious technical errors in the interpretation of the extensive electrochemical data, both in the text and SI, as detailed below.

Issues with electrochemical data and analysis:

1) Magnitudes of currents in the different solvents vary widely. Current is, in general terms, dependent on square root of the scan rate, electrode area, concentration of reactant and square root of the diffusion coefficient.

Presumably the area (1 mm Pt disc) was constant from experiment to experiment and we can compare equal scan rates. The concentrations are nominally constant (10 mM). The diffusion coefficient can be approximated by the Stokes Einstein equation

$$D = kT/(6 \pi \eta r_0)$$

so will vary with T (not specified in the manuscript), hydrodynamic radius of the species, and the solvent viscosity. Table 1 shows the solvent viscosity increases monotonically from 0.57 to 0.90 mPa s with the degree of fluorination.

It is therefore difficult to account for the apparently random variations of up to a factor of 2, in the currents in Figure 3. The same random variation is seen in the voltammetry data presented in the SI even under nominally identical conditions (Fig S2 and S3, Figure S4 and S5).

2) I do not understand why the authors choose to use 2nd cycle in the voltammetry in each case. The theory assumes a uniform concentration (no concentration polarisation) in solution and zero initial current (Nicholson and Shain, *Anal. Chem.* 1964, 36, 706 and references therein) and this only applies on the first cycle.

3) The linear variation of peak current with scan rate (page 7 and SI 3.4.1.1 to 3.4.4) is not a valid test for reversibility. The peak current also varies linearly with scan rate in the irreversible case and is close to linear with scan rate in the quasi reversible case (see, for example, Yamada, *Electrochemistry*, 90, 102005 (2022)). The variation of peak current with scan rate is diagnostic of a process involving diffusion of reactant and product in solution as opposed to surface adsorbed or surface bound species.

The key criteria for reversible voltammetry are a peak separation of $59/n$ mV at room temperature, a potential separation of the half peak and peak potential of $59/n$ mV at room temperature, no change in peak positions with scan rate, and equal anodic and cathodic peak heights (when corrected measured). Clearly the majority (if not all) the voltammograms in figure 3 and in the SI do not meet these criteria.

4) The peak separation in the different voltammograms varies significantly from case to case and the peak positions shift with scan rate both for the voltammograms in Figure 3 and for the voltammograms in the SI (for example Figure S2 and S3, and Figure S4 and S5). This suggests either significant distortions from iR drop and/or slow electron transfer kinetics (possibly caused by electrode fouling).

Assuming for the moment that this is caused by iR drop then the value of the uncompensated solution resistance must vary somewhat randomly from experiment to experiment, possibly due to variations in relative placement of the electrodes.

Taking the results in Figure S2 as an example. Assuming that the shifts in peak position are due to iR drop we can estimate the uncompensated solution resistance. This gives a value of around $5 \text{ k}\Omega$ (from the slope of line drawn through the anodic or cathodic peak currents). For the cycle at 200 mV/s the peak currents are around $+28$ and $-21 \text{ }\mu\text{A}$. These correspond to iR drops of $+140 \text{ mV}$ and 105 mV . This corresponds to a shift of $E_{1/2}$ of $+35 \text{ mV}$ caused by iR drop. (Note: because both peak currents vary with scan rate the effect of iR drop is not significantly dependent on scan rate. At 20 mV/s in Figure S2 the peak currents are around $+12.5$ and $-5 \text{ }\mu\text{A}$, corresponding to a $+37.5 \text{ mV}$ shift in $E_{1/2}$)

As a consequence if there is significant iR drop $E_{1/2}$ is not equal to E_0' as assumed on page 6 and 3.2.1 in the SI.

Comparing the shift calculated assuming iR drop from Figure S2 and the values in Table S3 we see that they agree. Further in those cases (such as Figure S8 for 5FB) where there appears to be significant iR drop there is a significant offset +ve potential offset in Table S3. (The one exception is for Figure S5 where the shift is in the wrong direction to be explained by IR drop.)

This same behaviour is seen for all of the voltammetric results reported in on pages SI 14 to SI 73 and this affects the interpretation of all of the electrochemical data and in turn the interpretation of all the results (for example in SI Section 3.5.3 or Table 3).

Other points:

Figure S1 appears to show a Pt gauze counter electrode clamped directly into a metal (brass?) holder. How is the solution prevented from contacting the brass and contaminating the solution with dissolution products from the brass?

In manuscript reference 106 is missing.

In the SI the references given stop at 101 but in the text go to at least 181.

Figure S26 is confusing. Is the shading in the lower section incorrect?

On pages 40 and 65 of the SI the authors refer to “Magic Blue”. I believe that Magic Blue is tris(4-bromophenyl)ammoniumyl hexachloroantimonate. If so the compound tris(4-bromophenyl)amine), TBPA, referred to here is not Magic Blue and the text is misleading. The voltammetry shown is not the voltammetry of MagicBlue it is the voltammetry of TBPA. I would expect the voltammetry of Magic Blue to be very different since it is a strong oxidant and there will also be voltammetry of chloride ion and of the antimony complex(es) present.

Minor typographical issues:

Chapter should be Section (instances on SI 7 and 9).

Figure S1 legend “glas” should be “glass”

Reviewer #2 (Remarks to the Author):

see attached file

In their manuscript “Pushing the redox potentials of deelectronators to highly positive values using solvent effects and weakly coordinating anions” the authors present a systematic study of increasingly fluorinated fluorobenzenes (xFB) and their effect as solvents on the redox potentials of NO^+ , Ag^+ , anthracene^F, and phenanthrene^F in combination with very potent WCAs like their well established perfluorinated tert-butoxy aluminate species. They differentiate non-innocent (NO^+ , Ag^+) from innocent (anthracene^F, phenanthrene^F) “deelectronators”, as they call oxidising agents. Identifying the right solvent leads to deelectronator-WCA pairs with remarkable redox potentials. They undertake extensive electrochemical examinations and analyse the thermodynamic properties of the formation reactions of various complexes and ion pairs employing quantum-chemical methods.

I think that overall this study is interesting, elaborate, and scientifically sound. A few inconsistencies and perhaps some more critical aspects can be found that I have detailed down below. In general, however, I am not sure about the novelty of this work. The authors say that the “progress on new [...] solvents almost stagnated”. However, they do not provide “new” solvents here, rather they look at solvents which the authors have already employed in previous works (cited in the paper) and analyse them in great detail. The authors should address this, before I can recommend this paper for publication.

General remarks

There is an issue with the references in both the manuscript as well as the Supporting Information, which made checking the literature quite bothersome. This needs to be addressed very carefully and should have been done prior to submission!

I find the term “de/electronator” rather weird, although I do not mind it being used in the paper. I see the authors’ reasoning, though I do not think that “deelectronation” is necessarily better than “oxidation”. In my opinion, these are just terms that students have to learn. Furthermore, the term “oxidation” is still found in the Supporting Information. I suggest the authors be absolutely consistent with this, if they want to push for a general change in the community, which I guess is the goal.

What is the significance of NO^+ and Ag^+ ? Is it just that they are simple/commercially available? Are these really the only “accessible and reliable reagents” with comparable redox potentials?

In the abstract it says “by +0.85 / +0.40 V higher than any published value.” This is technically true, I suppose. However, the authors’ own results suggest that DCM, if accompanied by an adequate WCA (pf⁻), pushes the potential of NO^+ to 1.4 V already. So, at least for NO^+ , it seems like the important point is to use an appropriate WCA, and not necessarily a special kind of solvent. However, the “revolutionary” aspect of this work is the systematic analysis of the solvent influence. I feel like this is a critical aspect to assess the novelty and relevance of the work. As the authors themselves say, there has been much work on WCAs in recent years already.

Further remarks

Manuscript

p.1 “Due to the weak solvent dependence of its potential, other electrochemical potentials can be compared for various liquid media (‘Ferrocene Assumption’).³ However, accessible and reliable reagents for deelectronation^{4,*} (= removal of an e^-) at potentials higher than about +0.5 to +1 V vs. Fc^+/Fc would be useful, but are scarce.” These two sentences seem detached. Why does it say “however”?

Fig. 1: The actual isovalue of the electrostatic potential plots should be provided.

p. 6 "... show that the [pf]⁻ salt gives a by about 1 V wider ECW than the [PF₆]⁻ salt". Sentences like these are found in the text from time to time. The sentence structure makes it a little hard to read. I would suggest to rearrange the sentence a little, e.g., "... show that the [pf]⁻ salt gives an ECW which is 1 V wider than that of the [PF₆]⁻ salt".

p. 6 "Hence, the basic electrochemical performance of the xFB solvents with [NBu₄]⁺[pf]⁻ as supporting electrolyte salt is very promising." I am not sure if this formulation is adequate. It seems like xFB solvents display standard ECWs in the general context. Could the authors comment on that?

p. 9 The "scXRD NO⁺ Salts" subsection lacks references to compare bond lengths with other (Wheland) complexes.

p. 9/10 Do the authors have an idea why 2FB---NO⁺ could be crystallised, but not with 1FB or 3FB?

p. 10 "For residues stemming from 4-6FB solution, a weak ν_{NO} band was observed at 2338 cm⁻¹ (Figure 5B) and hence essentially at the same position as in neat NO⁺[pf]⁻." While I guess we can agree that there *should* be a weak band for the NO vibration, I cannot say that I agree that it is actually visible, at least not in Fig. 5B.

p. 10 "Due to the low intensity of the band ν_{NO} in non-coordinated NO⁺, this stretching band was invisible in the solution." This sentence is redundant. Why does the NO stretching in non-coordinated NO⁺ display a low IR intensity?

p. 11 "COSMO-RS" lacks a reference.

p. 13 I assume that by "uncomplexed NO⁺ in solution" the authors mean that apparently no Wheland complex is formed. Have the authors done a proper analysis of structural and electronic properties to assess the binding situation in the computed complexes, i.e., can for example [NO(2FB)]⁺ in the gas-phase be called a Wheland complex? I note this, because otherwise I would presume that the NO⁺---S interactions are more or less purely electrostatic which would beg the question to what extent "complexed NO⁺" is different from the NO⁺ that is treated in the COSMO-RS calculation by constructing a cavity of solvent molecules around it which also interact electrostatically with the solute. This may culminate in an unnecessarily fundamental discussion about the nature of chemical bonding, which is of course beyond what this work is trying to do. I would suggest that the authors do a representative analysis of the bonding situation within one or two of their Wheland complexes. This could, for example, be done by analysing the potential energy surface of dissociation of NO⁺ from the xFB solvent. Or an energy decomposition analysis could be done. Since the authors have used the DLPNO-CCSD(T) method, ORCA provides the means to perform a "local energy decomposition analysis".

p. 13 "In addition, also the magic blue salts [MB]⁺[WCA]⁻, appraised¹⁷ for their innocent behavior" - "Appraised" is probably supposed to be "praised"?

p. 15 "If these de⁺[WCA]⁻ reagents (de = Ag, NO) do react non-innocent by coordination" It should be "non-innocently".

p. 15 “we note that the currently rather high prizes of 3FB and 4FB can substantially fall, if used frequently in the community.” Surely, when there is a higher demand for a certain good, people will try to make it more accessible, so that it can be better commercialised. However, the first response of the market, when there is a higher demand but not more supply for something, is that the price will increase. I would urge the authors to rephrase this paragraph or omit it, especially since it is the last one of the conclusion. Currently, to me it sounds rather allusive and more like an attempt to artificially increase the importance of the study.

p. 16 Fig. 8 “Gibbs Enthalpy” The authors are a little too lenient with their vocabulary for my taste as all versions of “Gibbs energy”, “Gibbs free energy”, “free energy”, and “Gibbs enthalpy” are found throughout the paper. I would suggest to stick with “Gibbs energy” or “Gibbs free energy”. (The thermodynamically “most correct” term would be “free enthalpy”, which is, however, rarely used in literature.)

p. 17 “An extended search of the potential energy surface was performed” How was this search performed? What algorithm was used?

Supporting Information

A few spelling mistakes “the the neutral NO molecule” and sentence structure difficulties “Unfortunately, we found during the first CV measurements with $\text{Fc}^+[\text{Al}\{\text{OC}(\text{CF}_3)_3\}_4]^-$ as an internal reference, silver signals, using the bulk product of this reaction.” are found. I would, for example, change this sentence to “Unfortunately, when using the bulk product of this reaction, we found silver signals during the first CV measurements with $\text{Fc}^+[\text{Al}\{\text{OC}(\text{CF}_3)_3\}_4]^-$ as internal standard.”

Section 4.1 could use a little refurbishment.

Section 4.1.1: Is B3LYP used to benchmark BP86? I appreciate the authors’ effort to justify the method used for structural optimisations, and I do think their general approach is valid, however, this is not state-of-the-art. If they really want to convince the reader that BP86 yields sufficiently accurate structures, they should add a few methods for comparison. There are more up-to-date methods such as Stefan Grimme’s PBEh-3c or John Perdew’s r2SCAN method. Usually, it is a good idea to vary the amount of Hartree-Fock exchange by, e.g., including Becke’s B3LYP or Truhlar’s M06-2X functionals. I recommend that the authors perform at least one more calculation for benchmarking BP86, preferably something state-of-the-art like the PBEh-3c composite method.

Regarding the usage of COSMO-RS: While the computed values from the Born-Haber-Fajans cycles are surely convincing, could the authors comment on the structural change of the complexes when going from vacuum to solution? This aspect has been addressed in section 4.3 for two systems, but I would be interested in general. In my own studies of weakly bound systems I have found that for certain systems, comparing gas-phase and solution-phase structures from calculations with continuum models may sometimes lead to significant structural changes which may in turn artificially (de-)stabilise a compound in solution, not because this reflects reality, but because explicit solvent effects are missing (even with COSMO-RS), which do not play a role in vacuum, but in solution. This is, of course, irrelevant for a structure like ferrocene, but it might impact the ion pairs. I would like to exclude systematic errors in the computation of solvation energies here.

Something seems to be off with Table S30, the arrows are a little confusing, I had to re-check the data to see what corresponds to which reaction. I suggest to adjust the description so that it is clear that the data refer to the successive addition of solvent molecules to the complex.

The errors in the ionisation energy calculation figures (section 4.5) seem to be mixed up. For the calculated values in Fig. S99 I get an error of $141.8 - (164.8 - 651.9 - 25.6 + 42.7 + 799 - 189.6) = 2.4$ kJ/mol (although it says 1.6 kJ/mol), and in Fig. S100 I get an error of $(164.8 - 651.9 - 25.6 + 41.6 + 844.2 - 189.3) - 182.4 = 1.4$ kJ/mol (although it says 2.4 kJ/mol).

The authors should be consistent when writing “DLPNO-CCSD(T)”. Both capital and small letter versions are found in the Supporting Information. I would prefer it written in capital letters (also in the manuscript).

As a general remark, I would urge the authors to please provide cartesian coordinates as a separate simple ascii file that can be read in. Copy and pasting the data from the PDF file can be extremely inconvenient (at least it was in my case when trying to perform a few calculations of my own).

Reviewer #3 (Remarks to the Author):

I have been asked, primarily, to provide feedback on the soundness of the research reported, and am happy to do so. This paper fits within a larger theme of research from the P.I.'s group over the past several years (Refs 13-15, possibly others but the Author Lists are incomplete...) aimed at developing a Unified Potential Scale. This is very important research and of great potential significance, as the world decarbonizes and will become dependent on electrochemistry on a very huge scale. This paper contributes to the theme by addressing the solvents that may be used for electrochemistry, from a fundamental (rather than the typical engineering-applied approach that litters the electrochemistry literature, driven as the latter is by industrial interests). This is hugely important work. The five solvents, 1-5(FB) developed and evaluated here, along with the anions [pf]- and [al-f-al]- from the P.I.'s previous important contributions to lowest-coordinating anions, provide significant breakthroughs.

Silver, as an oxidant, is of course of extreme importance in advanced redox chemistry and the strong passivation of its oxidation potential with coordinating solvents (water, acetonitrile, etc) is well established. Ag⁺ as an oxidant is much more powerful in a less-coordinating solvent and is a very useful as the redox partner is inert Ag metal. We use this all the time in our work. Now that this power can be increased by changing to these new solvents, the practicality has been significantly extended.

Of much greater significance in this work, however, is the wide-ranging intellectual expansion of ideas, all of which is substantiated by detailed analyses and by impressive synthetic chemistry, supported by innovative measurements, some excellent but very challenging crystal structures, and a high-level of computational analysis. The 240 page Supporting Information is well organized and comprehensive, making the overall package a treasure trove of novel and exciting science. The work is sound, the analyses insightful and supported by the evidence, and there is full documentation to allow for repetition of the work. I strongly support publication of this work in Nature Communications.

I do have one serious caveat: advances in science require excellent communication, and here the proclivity of the author group to employ newly defined or rare terms is, in my view, extremely counterproductive. Here I am focused on this weird term, the 'deelectronator' (which in English should almost surely be hyphenated, de-electronator, but that is just part of the problem). It is most distressing that this weird term is not once DEFINED within the body of this work – readers need to go to Google or dig through the author's previous works to try and unearth the meaning of this highly non-obvious word, which, as mentioned, is grammatically doubtful. Either this term should be replaced by a common term that all can understand, or it MUST be both (1) defined and (2) justified, in each and every publication, until it finds resonance with the electrochemical and chemical community (or gets abandoned in the dust of time).

Evaluation of SC-XRD

There are 7 SC-XRD structures, almost all of them difficult structures with a lot of disorder. All the models that have been developed are excellent. This reviewer suspects that several (6, 7, maybe

10) have superlattices, which needs commenting on. The disorder models will suffice but if there is evidence from the image data, this should be mentioned. Superlattices from the weak perfluoro salts could be entirely expected, and others working trying follow-up work could benefit from the discussion. Small issues with the refinements are mentioned here:

Structure of 5 – Pca2(1). This structure seems fine, with a good disorder model (i.e. a very challenging structure, well executed). However, it does not seem to have been refined to completion – this may be from using an out-of-date version of SHELXL? (2018 is reported....)

Structure of 6 – a twinned structure with a $Z' = 2$ in P-1. Sure there is not a superlattice involved? Again, refinement does not seem to be converged, possibly due to same software issue.

Structure of 7 – P2(1)/n. Also not fully refined but this is from some minor issue with Ag1, and is OK. The odd disorder between O1_9 and O1_8 could be improved, e.g. by an ISOR restraint on O1_9, perhaps? Here too with $Z'2$ and disorder, a superlattice may be operational. No modelling of that is required, but some comments in the S.I. Experimental might be advised.

Structure of 8. – P2(1)/c. No issues here. Just a 'normal' disorder! Structure is a very nice NO+ pi adduct.

Structure of 9 – P2(1)/c. Disordered C1_6 needs attention. ISOR restraints may do the trick. Otherwise fine, and a good disorder model.

Structure of 10 – P-1. Disordered, with a good disorder model developed. The weights do not seem to have been optimized – perhaps a SHELX version thing too.

Structure of 11 – P2(1)/c. Ordered 1:2 salt. Good structures. Weights again an issue, same possible reason. Please check.

Other issues:

Refs. 11-14, 25, 27, 31, 33 have incomplete author lists, and there are more of these.

Point-by-Point Answers to the Referees' Comments:

Reviewer #1

comment by the reviewer

Magnitudes of currents in the different solvents vary widely. Current is, in general terms, dependent on square root of the scan rate, electrode area, concentration of reactant and square root of the diffusion coefficient. Presumably the area (1 mm Pt disc) was constant from experiment to experiment and we can compare equal scan rates. The concentrations are nominally constant (10 mM). The diffusion coefficient can be approximated by the Stokes Einstein equation $D = kT/(6 \pi \eta r_0)$ so will vary with T (not specified in the manuscript), hydrodynamic radius of the species, and the solvent viscosity. Table 1 shows the solvent viscosity increases monotonically from 0.57 to 0.90 mPa s with the degree of fluorination. It is therefore difficult to account for the apparently random variations of up to a factor of 2, in the currents in Figure 3. The same random variation is seen in the voltammetry data presented in the SI even under nominally identical conditions (Fig S2 and S3, Figure S4 and S5).

I do not understand why the authors choose to use 2nd cycle in the voltammetry in each case. The theory assumes a uniform concentration (no concentration polarization) in solution and zero initial current (Nicholson and Shain, Anal. Chem. 1964, 36, 706 and references therein) and this only applies on the first cycle.

The linear variation of peak current with scan rate (page 7 and SI 3.4.1.1 to 3.4.4) is not a valid test for reversibility. The peak current also varies linearly with scan rate in the irreversible case and is close to linear with scan rate in the quasi reversible case (see, for example, Yamada, Electrochemistry, 90, 102005 (2022)). The variation of peak current with scan rate is diagnostic of a process involving diffusion of reactant and product in solution as opposed to surface

responses and changes made by the authors

We determined the diffusion coefficients by means of the pulsed field gradient spin echo PFGSE-NMR technique. Since, Ferrocenium = Fc^+ is paramagnetic, we determined that of Cobaltocenium = Cc^+ , with the same anion used for all the measurements in the manuscript, i.e. the $[pf]^- = [Al(OR^F)_4]^-$ ($R^F = C(CF_3)_3$) counterion. We have determined the single crystal structures of both salts, and their volumes agree to within 0.6 % and hence, with the expectation of very similar behaviour of Fc^+ and Cc^+ , we used the diamagnetic Cc^+ as model ion to determine the diffusion constants D with the concentrations $c(Fc) = c(Cc^+[pf]^-) = 0.001 \text{ mol L}^{-1}$. The results of the measurement are given in Table C1 together with static dielectric constants ϵ_s and viscosities η of S.

Table C1: Diffusion coefficients D as obtained from DOSY-NMR measurements.

S	ϵ_s	$D(Cc^+)$ / $\text{cm}^2 \text{ s}^{-1}$	$D(Fc)$ / $\text{cm}^2 \text{ s}^{-1}$	η / $\text{mPa}\cdot\text{s}$
1FB	5.8	$8.4 \cdot 10^{-6}$	$1.73 \cdot 10^{-5}$	0.5662
2FB	14.2	$1.23 \cdot 10^{-5}$	$1.61 \cdot 10^{-5}$	0.6217
3FB	22.1	$1.15 \cdot 10^{-5}$	$1.46 \cdot 10^{-5}$	0.6962
4FB	12.6	$9.5 \cdot 10^{-6}$	$1.31 \cdot 10^{-5}$	0.7826
5FB	4.5	$5.62 \cdot 10^{-6}$	$1.3 \cdot 10^{-5}$	0.8165

One notes from the last two columns in the Table C1 that the D of neutral Fc are indeed linearly correlated with the viscosity.

However, against the expectation of the referee, the D of the charged Cc^+ species, and thus also the expected currents (!), are **rather a function of a complex mixture of the static dielectric constant (main) and of viscosity (minor)**. Cc^+ ions diffuse fastest in 2FB and 3FB and slowest in 1FB and 5FB.

Since all D were measured in the neat xFB solvent, i.e. without any supporting electrolyte, the presence of a tenfold excess of supporting electrolyte salt $NBu_4^+[pf]^-$ may cause changing ion pairing that further varies $D(Cc^+)$ especially in the lower polarity solvents.

From our point of view the observed currents can be explained with this behaviour. We will discuss this issue with some additional notes below.

We agree to the reviewers' comment. However, since our concern are thermodynamic data in contrast to kinetic or mechanistic data, this issue is of minor importance. As shown in a section below, the cycle number of the CV is essentially irrelevant to measure the thermodynamic potential (but may have impact on kinetic magnitudes that are out of our scope).

The reviewer is right with his comment. Indeed, all the measured CVs are electrochemically quasi-reversible. We considered this in the revision and now formulate unambiguously.

Note that the systems are electrochemically quasi-reversible, since the peak-to-peak separation of the waves in the CVs increases with the scan rate.

adsorbed or surface bound species. The key criteria for reversible voltammetry are a peak separation of $59/n$ mV at room temperature, a potential separation of the half peak and peak potential of $59/n$ mV at room temperature, no change in peak positions with scan rate, and equal anodic and cathodic peak heights (when corrected measured). Clearly the majority (if not all) the voltammograms in figure 3 and in the SI do not meet these criteria.

The peak separation in the different voltammograms varies significantly from case to case and the peak positions shift with scan rate both for the voltammograms in Figure 3 and for the voltammograms in the SI (for example Figure S2 and S3, and Figure S4 and S5). This suggest either significant distortions from iR drop and/or slow electron transfer kinetics (possibly caused by electrode fouling).

Assuming for the moment that this is caused by iR drop then the value of the uncompensated solution resistance must vary somewhat randomly from experiment to experiment, possibly due to variations in relative placement of the electrodes. Taking the results in Figure S2 as an example. Assuming that the shifts in peak position are due to iR drop we can estimate the uncompensated solution resistance. This gives a value of around 5 k Ω (from the slope of line drawn through the anodic or cathodic peak currents). For the cycle at 200 mV/s the peak currents are around +28 and -21 μ A. These correspond to iR drops of +140 mV and 105 mV. This corresponds to a shift of $E_{1/2}$ of +35 mV caused by iR drop. (Note: because both peak currents vary with scan rate the effect of iR drop is not significantly dependent on scan rate. At 20 mV/s in Figure S2 the peak currents are around +12.5 and -5 μ A, corresponding to a +37.5 mV shift in $E_{1/2}$) As a consequence if there is significant iR drop $E_{1/2}$ is not equal to E_0' as assumed on page 6 and 3.2.1 in the SI. Comparing the shift calculated assuming iR drop from Figure S2 and the values in Table S3 we see that they agree. Further in those cases (such as Figure S8 for 5FB) where there appears to be significant iR drop there is a significant offset +ve potential offset in Table S3. (The one exception is for Figure S5 where the shift in in the wrong direction to be explained by IR drop.) This same behaviour is seen for all of the voltammetric results reported in on pages SI 14 to SI 73 and this affects the interpretation of all of the electrochemical data and in turn the interpretation of all the results (for example in SI Section 3.5.3 or Table 3).

With the measuring instrument used (VMP3 from BioLogic), it is possible to determine the resistance between the working and the reference electrode before a (series of) measurement(s) and to compensate for it (Bio-Logic Application Note #29; <http://www.bio-logic.info>). We did this for all measurements shown (see Table C2; typical values $Re(Z)$ were determined using what the manufacturer calls the ZIR-technique, in the range of 1 – 2 k Ω , and in accordance with the manufacturer's recommendations, we compensated 85 % of the resistance, as otherwise unstable behaviour of the potentiostat could not be ruled out).

Table C2: $Re(Z)$ as obtained with the ZIR technique and resulting uncompensated resistance R_u .

Measuring series according to Supplemental Figure (in S)	$Re(Z)$ / Ω	$R_u = Re(Z) * 0.15$ / Ω
S2 (1FB)	1378	207
S3 (1FB)	n.a.	
S54 (1FB)	1137	170
S59 (1FB)	969	145
S33 (1FB)	997	150
S4 (2FB)	1417	213
S5 (2FB)	n.a.	
S55 (2FB)	1340	201
S34 (2FB)	1373	206
S60 (2FB)	1265	190
S6 (3FB)	1607	241
S56 (3FB)	1282	192
S35 (3FB)	1506	226
S61 (3FB)	1421	213
S7 (4FB)	n.a.	
S57 (4FB)	1469	220
S36 (4FB)	n.a.	
S62 (4FB)	1182	177
S8 (5FB)	1802	270
S58 (5FB)	2526	379
S37 (5FB)	1990	299
S63 (5FB)	2529	379

Thus, the majority of the observed peak separations, if not all, are caused by slow electrode kinetics as the reviewer pointed out in comment 3.

Additional notes to the remarks 1 – 4 of referee 1:

We fully appreciate the reviewer's comments, as they show that accuracy has to be considered. With CV simulations shown below (Figures C1 and C2), we demonstrate that the results obtained from the CVs given in the manuscript and in the SI do not deviate from the accuracy of ± 0.025 V as stated by us in the paragraph 3.2.2 "Validation with Born-Fajans-Haber-Cycles". We use as examples results for the solvents 3FB and 5FB (as these have drawn particular attention to the reviewer) to show that – although the comments are justified from a theoretical point of view – the current height, the cycle number and the uncompensated resistance do not have a significant impact on the potential values we have specified within the stated accuracy of ± 0.025 V. Note, that the chosen solvents 3FB and 5FB represent the extremes in terms of solvent dipole density = dielectric constants: The one for 3FB (22.1) is the highest and that of 5FB (4.5) is the lowest in the entire series included with Table C1.

Hence, we have performed a series of simulations of the CVs for both solvents 3FB and 5FB using parameters from Table C1 / C2 and other kinetic magnitudes (collected in Tables C3 and C4). The goal was on the one hand to fit the CV simulations to the experimental CV curves and, once achieved, to systematically vary the input parameters exemplarily to extreme values and thereby analyse the impact of the (unreasonable) parameter variation on the thermodynamic half-wave potential relevant for this article.

Figure C1a shows the simulation of the CV of Fc^+ in 3FB given in Figure 3, performed with the program DigiElch 7 (by ElchSoft, www.elchsoft.com) and with the experimental parameters from Tables C1 and C2. The thermodynamic parameter E° as well as the kinetic parameters α and k^0 were adjusted in order to align the curve progression with that of the experiment. The value of k^0 indicates quasi-reversible kinetics of the system, as suggested by the reviewer.

In Figure C1b the diffusion coefficient $D(\text{Fc}^+)$ was additionally adjusted to account for the measured current height. Indeed, a slightly larger $D(\text{Fc}^+)$ than that obtained from the NMR experiment must be used. However, the change of $D(\text{Fc}^+)$ obviously does not cause a change of the peak potentials.

Figure C1c was simulated with $R_u = 5000 \Omega$, a value, which obviously is nonsense. But even in this case our stated E° value of -1.14 V is correctly represented within the uncertainty level of 0.05 V .

The same is true using an unreasonably small rate constant k^0 (Figure C1d).

The results of the simulations with the parameters used for Figure C1a-d are collected in Table C3.

Figure C1: Measured (green) and simulated (black) CVs of Figures 3 and S56 (Fc^+ vs. Ag^+/Ag in 3FB, $\nu = 0.1 \text{ V s}^{-1}$). The first three cycles of the CV measurement and simulated.

Table C3: Simulation parameters used to prepare Figure C1a-d in 3FB. Cases a) to d) are explained in the text. The $D(\text{Fc}^+)$ value that was changed with respect to the measurements collected in Table C1 are marked in **bold**. Extreme example values for k^0 and R_u are marked in **bold and italic**.

Parameters used for simulation								
Case	α	k^0 $/ \text{cm s}^{-1}$	R_u $/ \Omega$	$D(\text{Fc}^+)$ $/ \text{cm}^2 \text{ s}^{-1}$	$D(\text{Fc})$ $/ \text{cm}^2 \text{ s}^{-1}$	E° $/ \text{V}$	stated $E^\circ = E_{1/2}$ $/ \text{V}$	simulated CV $E_{1/2} =$ $/ \text{V}$
a)	0.5	0.008	192	$1.15 \cdot 10^{-5}$	$1.45 \cdot 10^{-5}$	-1.144	-1.14	-1.143
b)	0.5	0.008	192	$2.9 \cdot 10^{-5}$	$1.45 \cdot 10^{-5}$	-1.133	-1.14	-1.144
c)	0.5	0.008	5000	$2.9 \cdot 10^{-5}$	$1.45 \cdot 10^{-5}$	-1.133	-1.14	-1.161
d)	0.5	0.0008	192	$2.9 \cdot 10^{-5}$	$1.45 \cdot 10^{-5}$	-1.133	-1.14	-1.148

The same procedure as defined above for 3FB was performed for the measurements with Fc^+ in 5FB given in Figure 3. Figure C2a-d includes the CV traces and comes to the same result.

Again, in going from a) with the experimental parameters from Tables C1 and C2, to b) where the diffusion coefficient had to be adjusted to match the current height in experiment and simulation (in this case it was diminished compared to the value obtained by NMR experiment). Next, c) and d) assume unrealistic values, but show little change of the desired quantity (see Table C4).

Figure C2: Measured (green) and simulated (black) CVs of Figures 3 and S58 (Fc^+ vs. Ag^+/Ag in 5FB, $\nu = 0.1 \text{ V s}^{-1}$). The first three cycles of the CV measurement are shown and simulated.

Table C4: Simulation parameters used to prepare Figure C2a-d in 5FB. Cases a) to d) are explained in the text. The $D(\text{Fc}^+)$ value that was changed with respect to the measurements collected in Table C1 is marked in **bold**. Extreme example values for k^0 and R_u are marked in **bold and italic**.

Case	Parameters used for simulation						stated $E^{\circ'} = E_{1/2}$ / V	simulated CV $E_{1/2} =$ / V
	α	k^0 / cm s^{-1}	R_u / Ω	$D(\text{Fc}^+)$ / $\text{cm}^2 \text{ s}^{-1}$	$D(\text{Fc})$ / $\text{cm}^2 \text{ s}^{-1}$	E° / V		
a)	0.5	0.00063	379	$5.62 \cdot 10^{-6}$	$1.3 \cdot 10^{-5}$	-1.395	-1.38	-1.389
b)	0.5	0.00063	379	$3.5 \cdot 10^{-6}$	$1.3 \cdot 10^{-5}$	-1.398	-1.38	-1.385
c)	0.5	0.00063	5000	$3.5 \cdot 10^{-6}$	$1.3 \cdot 10^{-5}$	-1.398	-1.38	-1.392
d)	0.5	0.000063	379	$3.5 \cdot 10^{-6}$	$1.3 \cdot 10^{-5}$	-1.398	-1.38	-1.389

In both cases, the difference for changing the temperature from 298.15 K to 293.15 K is so small that the graphical plots were omitted here. Hence, we do exclude temperature to have any reasonable influence on the CV measurements. We now specify the temperature in the manuscript as “room temperature”.

Further, in Figures C1 and C2 all three recorded cycles are shown. It can be seen that the $E_{1/2}$ in all cycles, including the first, are the same to within less than 5 mV and do not vary. Hence, within the stated uncertainty of $\pm 0.025 \text{ V}$ it is not relevant which of the cycles is used for extracting $E_{1/2}$.

Conclusion to the Comment by Referee 1: The intention of this section is to demonstrate that the issues addressed by the reviewer, although correct from a theoretical and conceptual point of view, are practically not relevant for the determination of the thermodynamic parameter $E^{\circ'}$ within our stated uncertainty. This should be evident from Figures C1a-d and C2a-d. The issues addressed by the referee will clearly influence kinetic magnitudes, but those are not relevant for our work. We only refer to the potential values of the measurements, which are unaffected within our error margin of $\pm 0.025 \text{ V}$ extracted from the thermodynamic triangular Born-Fajans-Haber cycles. In addition, we have used direct chemical reactions to verify the magnitude of the potentials, i.e. see the deelectronation reactions of Ag^+ with anthracene^{Hal} and anthracene^F in the innocent deelectronator section of the manuscript.

In addition, a comparison with literature data of the Fc⁺/Fc system in some solvents that were measured by N.G. Tsierkezos is given in Table C5 (*J. Solution Chem.* **2007**, 36, 289). The observed peak potential separations are in quite good agreement (AN and DMF) or even lower, indicating a faster ET kinetics (DCM). (Clearly, since Tsierkezos used a Ag/AgCl reference electrode the potential positions of the peaks and of $E_{1/2}$ cannot be compared.) We therefore consider the data we have provided to be accurate within the stated uncertainty of ± 0.025 V.

Table C5: Comparison with literature data of the Fc⁺/Fc system in some solvents measured by N.G. Tsierkezos.

		Tsierkezos at 298.15 K (0.1 mol L ⁻¹ [Bu ₄ N][PF ₆])	Our results at room temperature (Fig. 85, 87, 88)
Solvent	$\nu / V s^{-1}$	$\Delta E_p / V$	$\Delta E_p / V$
AN	0.02	0.076	0.075
	0.05	0.079	0.079
	0.1	0.087	0.082
DMF	0.02	0.077	0.077
	0.05	0.080	0.083
	0.1	0.086	0.091
DCM	0.02	0.144	0.100
	0.05	0.157	0.108
	0.1	0.213	0.120

Continuation of the direct answers to the points raised by referee 1:

Figure S1 appears to show a Pt gauze counter electrode clamped directly into a metal (brass?) holder. How is the solution prevented from contacting the brass and contaminating the solution with dissolution products from the brass?

The counter-electrode, consisting of a Pt gauze clamped directly into a brass holder, was passed through a septum with the copper rod attached to it (see left part of Figure S1a), and only the lower part of the Pt gauze was immersed in the measurement solution. The septum ensured that the electrode could not slip downwards, thus avoiding contamination by dissolved brass.

The schematic setups have been modified accordingly to illustrate that only the Pt gauze was immersed into the measurement solution.

In manuscript reference 106 is missing.

The missing reference was added to the corresponding reference table.

In the SI the references given stop at 101 but in the text go to at least 181.

There was a problem with the conversion of the Word-file (with CITAVI-references) to the PDF version. We had carefully checked everything in Word, but after conversion to PDF, this was not done. We apologise for this and now also the numbering of references in the SI references has been revised to include all 102 references as they appear in the SI text.

Figure S26 is confusing. Is the shading in the lower section incorrect?

We apologise for the confusion, the reviewer correctly identified that the colour gradient on the right side in the lower section was incorrect. The figure has been changed accordingly.

On pages 40 and 65 of the SI the authors refer to "Magic Blue". I believe that Magic Blue is tris(4-bromophenyl)ammoniumyl hexachloroantimonate. If so the compound tris(4-bromophenyl)amine, TBPA, referred to here is not Magic Blue and the text is misleading. The voltammetry shown is not the voltammetry of MagicBlue it is the voltammetry of TBPA. I would expect the voltammetry of Magic Blue to be very different since it is a strong oxidant and there will also be voltammetry of chloride ion and of the antimony complex(es) present.

We apologise for the misleading nomenclature and can follow your arguments and have therefore changed the name of the molecule to TBPA.

Chapter should be Section (instances on SI 7 and 9).

Changed.

Figure S1 legend "glas" should be "glass"

Changed.

Reviewer #2

comment by reviewer #2

There is an issue with the references in both the manuscript as well as the Supporting Information, which made checking the literature quite bothersome. This needs to be addressed very carefully and should have been done prior to submission!

I find the term “de/electronator” rather weird, although I do not mind it being used in the paper. I see the authors’ reasoning, though I do not think that “deelectronation” is necessarily better than “oxidation”. In my opinion, these are just terms that students have to learn. Furthermore, the term “oxidation” is still found in the Supporting Information. I suggest the authors be absolutely consistent with this, if they want to push for a general change in the community, which I guess is the goal.

What is the significance of NO⁺ and Ag⁺? Is it just that they are simple/commercially available? Are these really the only “accessible and reliable reagents” with comparable redox potentials?

In the abstract it says “by +0.85 / +0.40 V higher than any published value.” This is technically true, I suppose. However, the authors’ own results suggest that DCM, if accompanied by an adequate WCA (pf-), pushes the potential of NO⁺ to 1.4 V already. So, at least for NO⁺, it seems like the important point is to use an appropriate WCA, and not necessarily a special kind of solvent. However, the “revolutionary” aspect of this work is the systematic analysis of the solvent influence. I feel like this is a critical aspect to assess the novelty and relevance of the work. As the authors themselves say, there has been much work on WCAs in recent years already.

responses and changes made by the authors

We are very sorry and have commented on this a bit before. It was conversion error from Word to PDF.

Well, we see this different. And it is explicitly the pedagogic aspect that drives us to use electronation and deelectronation. The terms are self-explanatory and even under pressure a student immediately uses the correct term. To the corresponding authors 20 years’ experience in teaching 1st year chemistry, even as an experienced lecturer and when introducing the terms and using them, one hesitates a second and thinks: “Is it removal or addition of an electron?” This never happens applying acid-base theory, as one speaks – closely related – about protonation and deprotonation. Using the terms “electronation and deelectronation”, introduced already 1925 in a seminal paper published by Hamilton B. Kady and Robert Taft in Science (<https://www.science.org/doi/10.1126/science.62.1609.403>), works intuitively, without hesitating. Hence, we use them – now also in the S.I. and thanks for noting this – consistently for *Electron Transfer*, in which the elementary steps do hold a direction and are termed accordingly *electronation* and *deelectronation*. In addition, dependably formed expressions like *deelectronator* etc. are utilized.

We have cited several papers on this. Clearly Ag⁺ and NO⁺ are not the only reagents (and we did not say that), but they are widely accessible and frequently used. Especially for the material sciences, i.e. battery materials, the commercial NO⁺ salts are very often used to delithiate and deelectronate the cathode active materials, e.g.:

Synthetically, we often do see the use of silver or NO⁺ salts by many, but frequently with competing reactivity as addressed explicitly in the manuscript. In the cited voltage range, not many reagents exist else and that are also widely used, e.g. the Magic Blue salt as hexachloroantimonate, thianthrenium salts and often also ferrocenium salts. But the latter have substantially lower potential.

Hence, in terms of simple availability and use by many in the field, those are the **typical** reagents. And this is all we are saying: those are frequently used and available reagents for the purpose in a potential range above +0.5 V vs. Fc⁺⁰.

Yes, it is all about the use of the correct ingredients in this **entire chemical system**. Hence, to achieve chemical deelectronation at high potential in low dielectric solvents (including DCM) one needs the right WCA-counterion for both, reagent as well as supporting electrolyte salt.

And it is to IK’s long-standing experience not clear to the community that many of the WCAs heavily used in the field (explicitly [B(Ar^F)₄]⁻ Ar^F = 3,5-(CF₃)₂-C₆H₃ and [B(C₆F₅)₄]⁻) are both, incompatible as salts of the non-solvated Ag⁺ and NO⁺ ions and decomposing (that means: de⁺[B(Ar^F)₄]⁻ and de⁺[B(C₆F₅)₄]⁻ are non-existent for de = Ag, NO) and, although the NBu₄⁺ salts of both WCAs were heavily pushed as supporting electrolyte salts by Bill Geiger in a series of

papers between 2000 and 2012, it was shown from work by the Waldvogel group, that supporting electrolytes with both fluoroorganoborate WCAs decompose at “anodic limit potentials” of only +1.25 to 1.78 V vs. Fc⁺/Fc in AN in electrochemical yields (!) with formation of the respective fluorinated biphenyls.

This is explicitly cited in the manuscript. Our claim here is, and this is the very useful novelty that will encourage applications:

“If you do want to undertake deelectronation chemistry in low dielectric media, this is a chemical system. In this system, all ingredients determine the potentials that may be reached. With the xFB solvents you have in part new or at least very little explored options, but only if you choose the right WCA counterions for deelectronator and/or supporting electrolyte you may reach potentials up to 1.89 V with the approach described.”

We do think, it is this message and remedy described, which makes the manuscript useful for many. In addition, the for the first time reported dielectric data as well as several of the physical properties of the xFB solvents will encourage use of these chemical systems by many and the revised Ag⁺ and NO⁺ potentials as well as ECWs in xFB and standard solvent solution will be frequently cited as a reference (just like the famous Geiger-Connelly review).

p.1 “Due to the weak solvent dependence of its potential, other electrochemical potentials can be compared for various liquid media (‘Ferrocene Assumption’).³ However, accessible and reliable reagents for deelectronation^{4,*} (= removal of an e⁻) at potentials higher than about +0.5 to +1 V vs. Fc⁺/Fc would be useful, but are scarce.” These two sentences seem detached. Why does it say “however”?

You are right. We changed this: First with a line break to separate the context and then setting deelectronation into context. It reads now:

“...various liquid media (‘Ferrocene Assumption’).³

By contrast to the multitude of reagents for electronation down to the potential level of solvated electrons, accessible and reliable reagents for deelectronation...”

Fig. 1: The actual isovalue of the electrostatic potential plots should be provided.

Added to the manuscript: **(0.025 e⁻ B⁻³)**

p. 6 “... show that the [pf]⁻ salt gives a by about 1 V wider ECW than the [PF₆]⁻ salt”. Sentences like these are found in the text from time to time. The sentence structure makes it a little hard to read. I would suggest to rearrange the sentence a little, e.g., “... show that the [pf]⁻ salt gives an ECW which is 1 V wider than that of the [PF₆]⁻ salt”.

We agree, have scanned through the manuscript and changed accordingly.

Now it reads: **“show that the [pf]⁻ salt gives an ECW which is by about 1 V wider than that of the [PF₆]⁻ salt”**

p. 6 “Hence, the basic electrochemical performance of the xFB solvents with [NBu₄]⁺[pf]⁻ as supporting electrolyte salt is very promising.” I am not sure if this formulation is adequate. It seems like xFB solvents display standard ECWs in the general context. Could the authors comment on that?

Hm, this formulation is ambiguous, we agree and only realized by your statement. It is not only the absolute value of the ECW width and the E_{pos} value that matter, but also the capability to be compatible with the requirements of the reactive cations generated. Not only in terms of their very deelectronating high potentials, but more general also in terms of the high Lewis acidity you induce in many chemical systems by deelectronation. The capabilities of especially the higher fluorinated xFB solvents to act as very weak ligands/donors, is further supported by the QM-calculated thermodynamics: Although being polar, the xFB solvents are extremely bad ligands to both Ag⁺ and NO⁺ (and in addition to many other very reactive cations that we analysed over the last years).

Again: The description as chemical system with system requirements for deelectronation is very important here. Thus, acetonitrile AN might look like a perfect choice in terms of the ECW (7.68 V with our electrolyte) and E_{pos} (+4.59 V). Yet, many of the potentially interesting reactive cations to be generated either intermediately (e.g. for catalysis) or synthetically (e.g. our transition metal carbonyl cation like $[\text{Fe}(\text{CO})_5]^+$, would react with the solvent AN, typically by coordination, as it is a rather good ligand. Hence, it is rather the combination of ECW, E_{pos} and inferior ligand properties which make the xFB solvents very attractive and unique. In addition, the chlorinated solvents like DCM often do react with the reactive cation by chloride abstraction. By contrast, we never observed fluoride abstraction from the xFB solvents.

Hence, we reformulated:

“They are compatible with a wide range of electrochemical syntheses. Hence, together with the poor capacity of the xFB solvent molecules to serve as ligands (see also sections 3.3 and 3.4), the basic electrochemical performance of xFB with $[\text{NBu}_4]^+[\text{pf}]^-$ as supporting electrolyte salt is very promising.”

p. 9 The “scXRD NO+ Salts” subsection lacks references to compare bond lengths with other (Wheland) complexes.

The missing references were added.

p. 9/10 Do the authors have an idea why 2FB---NO+ could be crystallised, but not with 1FB or 3FB?

Evaporation of the 1FB, 2FB and 3FB solutions of $\text{NO}[\text{pf}]$ yielded dark powders/crystals, whereas the evaporation of the 4FB, 5FB and 6FB solutions only led to white powders. Indeed, we also obtained single-crystals of the $[\text{NO}(1\text{FB})]^+$ and $[\text{NO}(3\text{FB})]^+$. However, the crystallographic data quality was too poor for the publication of the structures. The systematic study of the scXRD structures of these Wheland complexes would have been anyway beyond the scope of this paper. The structure of the $[\text{NO}(2\text{FB})]^+$ was added as a proof-of-concept for such an electron deficient Wheland complex.

p. 10 “For residues stemming from 4–6FB solution, a weak νNO band was observed at 2338 cm^{-1} (Figure 5B) and hence essentially at the same position as in neat $\text{NO}+[\text{pf}]^-$.” While I guess we can agree that there should be a weak band for the NO vibration, I cannot say that I agree that it is actually visible, at least not in Fig. 5B

The Figure 5B was altered and the spectra stemming from dried $\text{NO}[\text{pf}]$ solutions in 4FB–6FB were multiplied by the factor 5, revealing the presence of the very weak vibration of the non-coordinated NO^+ .

The following was added to the description of the Figure:

The spectra from 4FB–6FB are multiplied by the factor 5, due to the low intensity of the NO band of non-coordinated NO^+ (calc. at: **28 km mol⁻¹**) in comparison to the intensities of the NO bands in the Wheland complexes (calc. at: **>1,000 km mol⁻¹**).

p. 10 “Due to the low intensity of the band νNO in non-coordinated NO^+ , this stretching band was invisible in the solution.” This sentence is redundant. Why does the NO stretching in non-coordinated NO^+ display a low IR intensity?

The very weak NO band was observable in the spectra of the dried solutions due to the higher concentration in comparison to the solution. The lower NO^+ concentration in solution did not allow this.

Regarding the second point: The non-coordinated NO^+ cation has nearly no dipole moment, therefore the stretching mode of the particle only barely changes its dipole moment, yielding a low IR activity. This point was taken up in the description of Figure 5, see above.

p. 11 “COSMO-RS” lacks a reference.

The corresponding references for COSMO-RS were added.

p. 13 I assume that by “uncomplexed NO⁺ in solution” the authors mean that apparently no Wheland complex is formed. Have the authors done a proper analysis of structural and electronic properties to assess the binding situation in the computed complexes, i.e., can for example [NO(2FB)]⁺ in the gas-phase be called a Wheland complex? I note this, because otherwise I would presume that the NO⁺---S interactions are more or less purely electrostatic which would beg the question to what extent

“complexed NO⁺” is different from the NO⁺ that is treated in the COSMO-RS calculation by constructing a cavity of solvent molecules around it which also interact electrostatically with the solute. This may culminate in an unnecessarily fundamental discussion about the nature of chemical bonding, which is of course beyond what this work is trying to do. I would suggest that the authors do a representative analysis of the bonding situation within one or two of their Wheland complexes. This could, for example, be done by analysing the potential energy surface of dissociation of NO⁺ from the xFB solvent. Or an energy decomposition analysis could be done. Since the authors have used the DLPNO-CCSD(T) method, ORCA provides the means to perform a “local energy decomposition analysis”.

p. 13 “In addition, also the magic blue salts [MB]⁺[WCA]⁻, appraised¹⁷ for their innocent behavior” - “Appraised” is probably supposed to be “praised”?

p. 15 “If these de⁺[WCA]⁻ reagents (de = Ag, NO) do react non-innocent by coordination” It should be “non-innocently”.

p. 15 “we note that the currently rather high prizes of 3FB and 4FB can substantially fall, if used frequently in the community.” Surely, when there is a higher demand for a certain good, people will try to make it more accessible, so that it can be better commercialised. However, the first response of the market, when there is a higher demand but not more supply for something, is that the price will increase. I would urge the authors to rephrase this paragraph or omit it, especially since it is the last one of the conclusion. Currently, to me it sounds rather allusive and more like an attempt to artificially increase the importance of the study.

p. 16 Fig. 8 “Gibbs Enthalpy” The authors are a little too lenient with their vocabulary for my taste as all versions of “Gibbs energy”, “Gibbs free energy”, “free energy”, and “Gibbs enthalpy” are found throughout the paper. I would suggest to stick with “Gibbs energy” or “Gibbs free energy”. (The thermodynamically “most correct” term would be “free enthalpy”, which is, however, rarely used in literature.)

p. 17 “An extended search of the potential energy surface was performed” How was this search performed? What algorithm was used?

We now performed an EDA-NOCV analysis on the [NO(arene)]⁺ complexes with arene = mesitylene, benzene, xFB (x = 1–6). We modified the Figure 5 and included a new paragraph explaining the EDA-NOCV results. The full results are included in the ESI in Table S40. This is included with the manuscript:

Additionally, we performed an energy decomposition analysis combined with the natural orbitals of chemical valence (EDA-NOCV). In line with the experimental observations and other DFT calculations, the total interaction energy (ΔE_{int}) between the NO⁺ fragment and the different xFB fragments gradually decreases with increasing degree of fluorination (**Fehler! Verweisquelle konnte nicht gefunden werden.**D). Interestingly, the decrease of the total interaction energy is not due to a decrease of the orbital interaction energies (ΔE_{orb}), but the electrostatic interaction (ΔE_{elstat}). The entire EDA-NOCV results also including analyses for benzene and mesitylene are included in **Fehler! Verweisquelle konnte nicht gefunden werden.**

Changed.

Changed.

We can follow your arguments and decided to delete the entire paragraph.

We decided to conclusively use Gibbs energy, as this is the most frequently used term in the English literature.

Maybe as a side and explaining why we sometimes did use Gibbs enthalpy: In the German literature you address G almost exclusively as “Freie Enthalpie” = Free Enthalpy. Hence, sometimes ‘enthalpy’ was used instead of ‘energy’.

This search was done manually with a multitude of starting structures, not automated by using computer code.

Hence, it reads now: “An extended search of the potential energy surface was **manually** performed ”

A few spelling mistakes “the the neutral NO molecule” and sentence structure difficulties “Unfortunately, we found during the first CV measurements with $\text{Fc}^+[\text{Al}\{\text{OC}(\text{CF}_3)_3\}_4]^-$ as an internal reference, silver signals, using the bulk product of this reaction.” are found. I would, for example, change this sentence to “Unfortunately, when using the bulk product of this reaction, we found silver signals during the first CV measurements with $\text{Fc}^+[\text{Al}\{\text{OC}(\text{CF}_3)_3\}_4]^-$ as internal standard.”

Section 4.1 could use a little refurbishment.

Section 4.1.1: Is B3LYP used to benchmark BP86? I appreciate the authors’ effort to justify the method used for structural optimisations, and I do think their general approach is valid, however, this is not state-of-the-art. If they really want to convince the reader that BP86 yields sufficiently accurate structures, they should add a few methods for comparison. There are more up-to-date methods such as Stefan Grimme’s PBEh-3c or John Perdew’s r2SCAN method. Usually, it is a good idea to vary the amount of Hartree-Fock exchange by, e.g., including Becke’s B3LYP or Truhlar’s M06-2X functionals.

I recommend that the authors perform at least one more calculation for benchmarking BP86, preferably something state-of-the-art like the PBEh-3c composite method.

Regarding the usage of COSMO-RS: While the computed values from the Born-Haber-Fajans cycles are surely convincing, could the authors comment on the structural change of the complexes when going from vacuum to solution? This aspect has been addressed in section 4.3 for two systems, but I would be interested in general. In my own studies of weakly bound systems I have found that for certain systems, comparing gas-phase and solution-phase structures from calculations with continuum models may sometimes lead to significant structural changes which may in turn artificially (de-)stabilise a compound in solution, not because this reflects reality, but because explicit solvent effects are missing (even with COSMO-RS), which do not play a role in vacuum, but in solution. This is, of course, irrelevant for a structure like ferrocene, but it might impact the ion pairs. I would like to exclude systematic errors in the computation of solvation energies here.

We apologise for the spelling and grammatical errors identified by the reviewer and have corrected and amended them accordingly.

No, this was misunderstood. The energetics were calculated using the DFT structures for single point energy calculations at the DLPNO-CCSD(T) level extrapolated to the basis set limit and the respective DFT calculation for the correction to enthalpic and entropic contributions. Hence, the comparison really relies on much better energetics, than the methods recommended by the referee. In addition, we have done this evaluation for quite some more molecules and hence decided to stick for the large PES evaluation to only rely on the structures and enthalpic / entropic contributions from the simpler (but very robust and reliable) BP86/def2-TZVPP method.

This is addressed in the Table S24 now:

Table S1: Summary of the comparison between the DFT functionals BP86 and B3LYP for the dissociation of $\{\text{Ag}[\text{pf}]\}_{\text{ip}}$ into the ions Ag^+ and $[\text{pf}]^-$.

Calculated thermodynamics ^{a)} →	$\Delta G_r^\circ / \text{kJ mol}^{-1}$	$\Delta H_r^\circ / \text{kJ mol}^{-1}$
↓ Structure obtained by DFT functional:		
BP86	413.60	452.39
B3LYP	412.45	452.29

^{a)} Calculated using either a BP86/def2-TZVPP or a B3LYP/def2-TZVPP structure as the basis for a series of single point calculation at the DLPNO-CCSD(T)/cc-pVQZ for the gas phase energies at 0 K in vacuum and enthalpic / entropic contributions at the respective DFT level to correct to standard conditions.

If done otherwise, the energy differences would not be that minimal and within one kJ mol^{-1} . But, this minimal difference of the energetics with structures that were obtained with rather different methods shows, that both are already in a very shallow area of the PES at the CCSD(T) level: otherwise the energy difference would have been much larger.

Well, this would be reflected in the structures optimized by COSMO and Turbomole. We actually have published on such subtle differences when working on our cluster continuum models that we have termed rCCC and vCCC (relaxed and vertical Cosmo Cluster Continuum models, see: <https://doi.org/10.1002/chem.201003164>, also here: <http://dx.doi.org/10.1002/chem.201405391>).

What we noted is that for the given compounds the structures themselves did not change too drastically, but the energetic levelling of the most favourable structures either in the gas phase or by including solvation with COSMO-RS does change the relative stability and ordering.

We had done the entire solvation work also using only the COSMO or CPCM Solvation Model using either Turbomole or ORCA. Our approach to shed light on the performance and extending the (as the referee correctly states) simpler, because structurally very robust structures used in the BFHCs in SI, section 4.5., we have evaluated the capacity of

both methods (COSMO/CPCM vs. COSMO RS) to reproduce the energetics and single crystal structures found for the very weakly bound silver complexes. Here we found that **only** by using COSMO-RS and calculating the enthalpies (not Gibbs energies, see SI, section 4.2.2.) we could reproduce the relative ordering of the structures prevailing in solution and the solid state (entropy does not play a role for the preferred solid-state structures). And, to come to the referee's point, this also did include the ion-pairs. And, especially for the calculation of these large ion-paired systems, also the basis-set dependence was largest and in part did change from triple to quadruple zeta at the DLPNO-CCSD(T) level by about 10 kJ mol⁻¹. This is finally, why we decided to make a basis set extrapolation to the complete basis set limit (see SI, section 4.2.1). This is as good as you can possibly go currently. These calculations did run forever, even on the excellent JUSTUS computer cluster used.

To sum up, we are aware of these effects and have done our very best that no systematic errors were introduced. The only case, where the calculations (regardless if CPCM or COSMO-RS) consistently overestimated the solvation energies, were for the NO⁺ case, which as a small and isolated ion was always (seemingly) preferred by the calculation, rather than the version in the Wheland complex. This is stated in the text.

Something seems to be off with Table S30, the arrows are a little confusing, I had to re-check the data to see what corresponds to which reaction. I suggest to adjust the description so that it is clear that the data refer to the successive addition of solvent molecules to the complex.

We added "successive" to the table caption and tried make the labelling clearer. A section of this is shown here:

$\{(S)Ag[\rho f]\}_{ip} \xleftarrow{+S}$	$\{Ag[\rho f]\}_{ip} \xleftarrow{+[pf]}$	Ag^+	$\xrightarrow{+S} [Ag(S)_1]^+$	$\xrightarrow{+S} [Ag(S)_2]^+$
-43.4 / +12.5	-442.0 / -403.2		-78.8 / -49.1	-80.5 / -42.5
-28.9 / +20.3	-140.4 / -116.7		-31.5 / -9.3	-49.6 / -18.9

The errors in the ionisation energy calculation figures (section 4.5) seem to be mixed up. For the calculated values in Fig. S99 I get an error of 141.8 – (164.8 – 651.9 – 25.6 + 42.7 + 799 – 189.6) = 2.4 kJ/mol (although it says 1.6 kJ/mol), and in Fig. S100 I get an error of (164.8 – 651.9 – 25.6 + 41.6 + 844.2 – 189.3) – 182.4 = 1.4 kJ/mol (although it says 2.4 kJ/mol).

Correct, somehow the values are mixed up. The calculations from the reviewer are correct and the manuscript was changed accordingly.

The authors should be consistent when writing "DLPNO-CCSD(T)". Both capital and small letter versions are found in the Supporting Information. I would prefer it written in capital letters (also in the manuscript).

We have standardised the spelling of DLPNO-CCSD(T) in accordance with the reviewer's suggestion.

As a general remark, I would urge the authors to please provide cartesian coordinates as a separate simple ascii file that can be read in. Copy and pasting the data from the PDF file can be extremely inconvenient (at least it was in my case when trying to perform a few calculations of my own).

Hm, I guess everything is very properly included with this very long section of the S.I. IK very often complains himself as a referee that authors are not giving full detail here, but this is definitely not the case. I have arbitrarily copied one ion out of the many from the S.I. and include it here below. In addition, all single point DLPNO-CCSD(T) energies with DZ, TZ and QZ basis are included with Table S28 and those extrapolated to the basis set in Table S25. To our understanding this is very transparent and follows all instructions for authors of the Nature publication series.

[Ag(THF)₁]⁺

Method: (RI-)BP86(D3BJ)/def2-TZVPP

Symmetry: c1

Cartesian coordinates in Ångström:

C	-5.3473226	3.7148061	0.0294314
O	-5.8972081	2.3398012	-0.1278042
C	-4.9357063	1.3417226	0.4004499
C	-3.6213339	2.0969703	0.4383312
C	-4.0604673	3.5232289	0.8166115
H	-3.3107075	4.2757374	0.5481432
H	-4.2499589	3.5962881	1.8952055
H	-6.0997147	4.3217594	0.5453235
H	-5.1801070	4.0986581	-0.9847549
H	-4.9558740	0.4894847	-0.2879238
H	-5.2744163	1.0350089	1.3993752
H	-2.9293283	1.6664794	1.1709600
H	-3.1378685	2.0817194	-0.5468277
Ag	-7.9487165	1.9161156	-0.5130809

SCF energy GEOOPT = -379.4366163956 H

ZPE = 301.0 kJ/mol

FREEH energy = 319.65 kJ/mol

FREEH entropy = 0.36833 kJ/mol/K

\$vibrational spectrum

# mode	symmetry	wave number	IR intensity	selection	
rules					
#		cm**(-1)	km/mol	IR	
RAMAN					
1		-0.00	0.00000	-	-
2		-0.00	0.00000	-	-
3		0.00	0.00000	-	-
4		0.00	0.00000	-	-
5		0.00	0.00000	-	-
6		0.00	0.00000	-	-
7	a	48.30	2.46033	YES	YES
8	a	89.64	5.00228	YES	YES
9	a	116.33	0.60381	YES	YES
10	a	206.58	4.19072	YES	YES
11	a	248.40	0.17942	YES	YES
12	a	551.85	1.90396	YES	YES
13	a	667.59	0.53366	YES	YES
14	a	809.17	37.72534	YES	YES
15	a	817.21	98.75803	YES	YES
16	a	831.92	3.63697	YES	YES
17	a	901.33	8.60829	YES	YES
18	a	908.38	2.11871	YES	YES
19	a	944.70	25.50079	YES	YES
20	a	973.67	40.45575	YES	YES
21	a	1030.48	2.34773	YES	YES
22	a	1119.77	5.15082	YES	YES
23	a	1136.69	3.01390	YES	YES
24	a	1145.76	2.52656	YES	YES
25	a	1213.65	0.76187	YES	YES
26	a	1235.60	10.81246	YES	YES
27	a	1285.73	0.54666	YES	YES
28	a	1313.37	0.12772	YES	YES
29	a	1330.74	3.69760	YES	YES
30	a	1345.80	9.07870	YES	YES
31	a	1439.95	7.93721	YES	YES
32	a	1445.77	8.96909	YES	YES
33	a	1464.67	1.51144	YES	YES
34	a	1474.88	6.34646	YES	YES
35	a	2986.98	17.72328	YES	YES
36	a	2997.76	13.40670	YES	YES
37	a	3001.53	1.82174	YES	YES
38	a	3007.62	8.08847	YES	YES
39	a	3052.21	4.00719	YES	YES
40	a	3053.82	3.26240	YES	YES
41	a	3062.94	0.03280	YES	YES
42	a	3070.42	11.06102	YES	YES

\$end

Reviewer #3

comment by reviewer #3

I do have one serious caveat: advances in science require excellent communication, and here the proclivity of the author group to employ newly defined or rare terms is, in my view, extremely counterproductive. Here I am focused on this weird term, the 'deelectronator' (which in English should almost surely be hyphenated, de-electronator, but that is just part of the problem). It is most distressing that this weird term is not once DEFINED within the body of this work – readers need to go to Google or dig through the author's previous works to try and unearth the meaning of this highly non-obvious word, which, as mentioned, is grammatically doubtful. Either this term should be replaced by a common term that all can understand, or it MUST be both (1) defined and (2) justified, in each and every publication, until it finds resonance with the electrochemical and chemical community (or gets abandoned in the dust of time).

Structure of 5 – Pca2(1). This structure seems fine, with a good disorder model (i.e. a very challenging structure, well executed). However, it does not seem to have been refined to completion – this may be from using an out-of-date version of SHELXL? (2018 is reported....)

Structure of 6 – a twinned structure with a $Z' = 2$ in P-1. Sure there is not a superlattice involved? Again, refinement does not seem to be converged, possibly due to same software issue.

Structure of 7 – P2(1)/n. Also not fully refined but this is from some minor issue with Ag1, and is OK. The odd disorder between O1_9 and O1_8 could be improved, e.g. by an ISOR restraint on O1_9, perhaps? Here too with $Z'2$

responses and changes made by the authors

We are sorry that the referee feels offended by the use of electronation and deelectronation and depending terms that include the word stem.

Moreover, it has been defined at the first occasion, when used, i.e. on p. 2 top:

“accessible and reliable reagents for deelectronation^{4,*} (= removal of an e⁻) at potentials higher“

Including the footnote *:

“* Note, that we use electronation⁴ and deelectronation in their strict sense, i.e. addition / removal of electrons in innocent reactions, hence without non-innocent complications such as complex formation, substitution or degradation reactions. For reasoning see ref. 5 and the literature included with ref. 6.”

In addition, all the context given above to referee 1 holds.

May be only to add that in addition to the 1925 Science citation also the textbook *Modern Electrochemistry*, Plenum Press, New York, 1970 by J.O.M. Bockris and A.K.N. Reddy adopted this terminology, which, based on their use, found entry into the IUPAC gold book [doi: 10.1351/goldbook.E01978 as well as doi: 10.1351/goldbook.D01551].

We know that it is very difficult to change opinions on scientific vocabulary. But we are simply convinced that oxidation and reduction are the wrong expressions that are by no means intuitively to apply. By contrast, electronation and deelectronation are intuitively used correctly.

And if the term is hyphenated or not is matter of British or American English (BE or AE). Typically, we stick to AE, since IK did spend his postdoc period in an environment where AE was used. Yet, we are happy to change for Nature that is printed in BE.

Thank you for this comment! We indeed used an out-of-date version of SHELXL from 2018 for the last years. We use from now on the newer version from 2019. Unfortunately, there is no newer version. The difference from the version of the version from 2018 and 2019 are rather small and the parameters barely change or not even at all. All the bond distances stay the same. Therefore, we do not revise all of our structures, as the version of 2018 was also a valid one. We however use the version of 2019 for the revision of the structures, which have other issues.

No hints on a possible superlattice was found in the reciprocal lattice.

The structure was revised and the minor issue was solved with SIMU and RIGU restraints instead. This structure features indeed a superlattice based on weak hints in the reciprocal lattice along the a-axis. This is verified in the

and disorder, a superlattice may be operational. No modelling of that is required, but some comments in the S.I. Experimental might be advised.

packing of the model, wherein the silver atoms are nearly perfectly eclipsed to one another, when looked along the a-axis. It was commented in the SI:

^a Rows of reflexes with alternating intensity were observed along the a*-axis. Both the intense and less intense reflexes were considered resulting in a larger cell (supercell). The molecular moieties in the asymmetric unit differ slightly in their conformation.

Structure of 8. – P2(1)/c. No issues here. Just a 'normal' disorder! Structure is a very nice NO+ pi adduct.

Thank you.

Structure of 9 – P2(1)/c. Disordered C1_6 needs attention. ISOR restraints may do the trick. Otherwise fine, and a good disorder model.

Thank you. The structure was revised and the minor issue was solved with SIMU and RIGU restraints instead.

Structure of 10 – P-1. Disordered, with a good disorder model developed. The weights do not seem to have been optimized – perhaps a SHELX version thing too.

Thank you. See comment above

Structure of 11 – P2(1)/c. Ordered 1:2 salt. Good structures. Weights again an issue, same possible reason. Please check.

Thank you. See comment above

REVIEWERS' COMMENTS

Reviewer #1 (Remarks to the Author):

The authors have responded to the comments from the referees and revised the manuscript. I am satisfied by the authors' responses and happy for the revised manuscript to be published.

As an aside, in response to the authors' response to the points made about the voltammetry (referee 1), it is interesting that the peak currents vary significantly between 3FB and 5FB in Figure 3. As the authors' say, this seems to be related to the differences in dielectric constant (Table C1) and thus probably to ion pairing. The difference in peak current implies approximately a factor of 10 difference in diffusion coefficient in the two solvents ($i(\text{peak})$ is proportional to \sqrt{D}). The corresponding difference in electrode kinetics between 3FB and 5FB is also striking. It would be interesting to investigate these effects further.

Reviewer #2 (Remarks to the Author):

The authors have adequately addressed my questions and concerns, although I was a little confused as to why it was necessary to respond to my remarks with "Well" and "Hm".

I have just two comments in re-response:

1) Concerning the de/electronation terminology:

There is no question about the pedagogic advantage of the term, the authors are surely right here - and like I said, I do not mind it being used in the paper.

However, I do feel that I have to give my support to reviewer #3's point about communication. The authors claim that they "know that it is very difficult to change opinions on scientific vocabulary", but refuse to adjust the footnote that they are referencing so that the terminology is more clearly justified.

I am sure that everyone agrees that this debate has little value, if it were only to result in scholars arguing about semantic notions. I therefore recommend to use pedagogic arguments in the main text and to justify it as referee #3 suggested. A quick search on Google yielded this paper:

Educ. Sci. 2020, 10(7), 170; <https://doi.org/10.3390/educsci10070170>.

In section 2 you can, for example, find the phrase "[t]his use of several models can be disconcerting for some students because it focuses on alternative definitions, such as applying the oxygen-based definition of redox reactions to identify all redox processes", which, in my opinion, may be a helpful thing to point out in the context of the de/electronation terms and your paper's introduction.

2) Concerning the provision of cartesian coordinates:

I did not argue that the authors' presentation was not transparent enough - quite the opposite, I think the presentation is absolutely fine. I merely think that providing a separate file with cartesian coordinates for the calculated structures is much more convenient for anybody who wants to re-do your calculations.

I fear that the issue can be platform-related (Windows v. Mac v. Linux). Copying xyz data from a PDF file may be simple on one machine, but bothersome on another (e.g., one row is turned into multiple after pasting). With a plain ASCII file this is never the case.

Reviewer #3 (Remarks to the Author):

In this revision to a previously submitted and reviewed version, Krossing and co-authors provide carefully corrected article and supporting information documents. I feel that all the substantive points raised by three very thorough and insightful reviews have been adequately addressed. Indeed, some of the more subtle points such as the previously inexplicable variations in currents in otherwise similar voltammetric measurements have been further investigated with some very interesting results.

All in all, this is a very major investigation, which has potential significance for changing paradigms and the enabling of significant new science, with relevance to important developments in the all-important applications of electrochemistry to energy storage and de-carbonization. Since all important issues in the reviews have been adequately addressed, I recommend publication of the revised version without reservations.

Point-by-Point Answers to the Referees' Comments:

Reviewer #1

comment by the reviewer	responses and changes made by the authors
As an aside, in response to the authors' response to the points made about the voltammetry (referee 1), it is interesting that the peak currents vary significantly between 3FB and 5FB in Figure 3. As the authors' say, this seems to be related to the differences in dielectric constant (Table C1) and thus probably to ion pairing. The difference in peak current implies approximately a factor of 10 difference in diffusion coefficient in the two solvents ($i(\text{peak})$ is proportional to \sqrt{D}). The corresponding difference in electrode kinetics between 3FB and 5FB is also striking. It would be interesting to investigate these effects further.	The authors fully agree and we plan to investigate this further in the future.

Reviewer #2

comment by reviewer #2	responses and changes made by the authors
The authors have adequately addressed my questions and concerns, although I was a little confused as to why it was necessary to respond to my remarks with "Well" and "Hm".	The authors are not native English speakers. A negative connotation was never intended with that. We apologize for that.
Concerning the de/electronation terminology: There is no question about the pedagogic advantage of the term, the authors are surely right here - and like I said, I do not mind it being used in the paper. However, I do feel that I have to give my support to reviewer #3's point about communication. The authors claim that they "know that it is very difficult to change opinions on scientific vocabulary", but refuse to adjust the footnote that they are referencing so that the terminology is more clearly justified. I am sure that everyone agrees that this debate has little value, if it were only to result in scholars arguing about semantic notions. I therefore recommend to use pedagogic arguments in the main text and to justify it as referee #3 suggested. A quick search on Google yielded this paper: Educ. Sci. 2020, 10(7), 170; https://doi.org/10.3390/educsci10070170. In section 2 you can, for example, find the phrase "[t]his use of several models can be disconcerting for some students because it focuses on alternative definitions, such as applying the oxygen-based definition of redox	Unfortunately, Nature Communications does not allow footnotes, therefore the terms would have to be discussed in the main text. Since the manuscript is too long anyway, according to the editors' comment, this is unfortunately outside of the scope of the manuscript, as no other oxidation reactions are discussed apart from the deelectronation. Nevertheless, the authors thank the reviewer for the valuable reference (s)he provided, as the authors plan to submit a paper just concerning this point by the end of the year. I.e., suggestions to the precise use of this terminology.

reactions to identify all redox processes", which, in my opinion, may be a helpful thing to point out in the context of the de/electronation terms and your paper's introduction.

Concerning the provision of cartesian coordinates: I did not argue that the authors' presentation was not transparent enough - quite the opposite, I think the presentation is absolutely fine. I merely think that providing a separate file with cartesian coordinates for the calculated structures is much more convenient for anybody who wants to re-do your calculations. I fear that the issue can be platform-related (Windows v. Mac v. Linux). Copying xyz data from a PDF file may be simple on one machine, but bothersome on another (e.g., one row is turned into multiple after pasting). With a plain ASCII file this is never the case.

A .txt file with all the coordinates is supplied with the revision. The editorial board raised a similar point.

Reviewer #3

comment by reviewer #3

responses and changes made by the authors

-

-